# Probabilistic Stability Guarantees for Feature Attributions

**Helen Jin**[*]
University of Pennsylvania
helenjin@seas.upenn.edu

**Anton Xue**[*]
University of Texas at Austin
anton.xue@austin.utexas.edu

**Weiqiu You**
University of Pennsylvania
weiqiuy@seas.upenn.edu

**Surbhi Goel**
University of Pennsylvania
surbhig@seas.upenn.edu

**Eric Wong**
University of Pennsylvania
exwong@seas.upenn.edu

## Abstract

Stability guarantees have emerged as a principled way to evaluate feature attributions, but existing certification methods rely on heavily smoothed classifiers and often produce conservative guarantees. To address these limitations, we introduce soft stability and propose a simple, model-agnostic, sample-efficient stability certification algorithm (SCA) that yields non-trivial and interpretable guarantees for any attribution method. Moreover, we show that mild smoothing achieves a more favorable trade-off between accuracy and stability, avoiding the aggressive compromises made in prior certification methods. To explain this behavior, we use Boolean function analysis to derive a novel characterization of stability under smoothing. We evaluate SCA on vision and language tasks and demonstrate the effectiveness of soft stability in measuring the robustness of explanation methods.

## 1  Introduction

Powerful machine learning models are increasingly deployed in practice. However, their opacity presents a major challenge when adopted in high-stakes domains, where transparent explanations are needed in decision-making. In healthcare, for instance, doctors require insights into the diagnostic steps to trust a model and effectively integrate it into clinical practice [32]. In the legal domain, attorneys must likewise ensure that model-assisted decisions meet stringent judicial standards [53].

There is much interest in explaining the behavior of complex models. One popular class of explanation methods is *feature attributions* [39, 51], which aim to select the input features most important to a model's prediction. However, many explanations are *unstable*, such as in Figure 1, where additionally including a few features may change the output. Such instability suggests that the explanation may be unreliable [47, 65, 72]. This phenomenon has motivated efforts to quantify how model predictions vary with explanations, including the effects of adding or removing features [55, 68] and the influence of the selection's shape [23, 54]. However, most existing works focus on empirical measures [3], with limited formal guarantees of robustness.

To address this gap, prior work in Xue et al. [70] considers stability as a formal certification framework for robust explanations. In particular, a *hard stable* explanation is one where adding any small number of features, up to some maximum tolerance, does not alter the prediction. However, finding this tolerance is non-trivial: for an arbitrary model, one must exhaustively enumerate and check all possible perturbations in a computationally intractable manner. To overcome this, Xue et al. [70] introduce the MuS algorithmic framework for constructing smoothed models, which have mathematical properties

---

[*]⟳ Equal contribution. Code is available at: https://github.com/helenjin/soft_stability/

39th Conference on Neural Information Processing Systems (NeurIPS 2025).

| Original Image | Explanation | +3 Features |
|:---:|:---:|:---:|
| 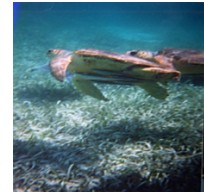 | 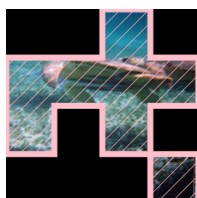 | |
| Loggerhead Sea Turtle ✓ | Loggerhead Sea Turtle ✓ | Coral Reef ✗ |

Figure 1: **An unstable explanation.** Given an input image (left), the LIME explanation method [51] identifies features (middle, in pink) that preserve Vision Transformer's [17] prediction. However, this explanation is not stable: adding just three more features (right, in yellow) flips the predictions.

for efficiently and non-trivially lower-bounding the maximum tolerance. While this is a first step towards certifiably robust explanations, it yields conservative guarantees and relies on smoothing.

In this work, we introduce *soft stability*, a new form of stability with mathematical and algorithmic benefits over hard stability. As illustrated in Figure 2, hard stability certifies whether all small perturbations to an explanation yield the same prediction, whereas soft stability quantifies how often the prediction is maintained. Soft stability may thus be interpreted as a probabilistic relaxation of hard stability, which enables a more fine-grained analysis of explanation robustness. Crucially, this shift in perspective allows for model-agnostic applicability and admits efficient certification algorithms that provide stronger guarantees. This work advances our understanding of robust feature-based explanations, and we summarize our contributions below.

**Soft stability is practical and certifiable**    To address the limitations of hard stability, we introduce soft stability as a more practical and informative alternative property in Section 2. Its key metric, the stability rate, provides a fine-grained characterization of robustness across perturbation radii. Unlike hard stability, soft stability yields non-vacuous guarantees even at larger perturbations and enables meaningful comparisons across different explanation methods.

**Sampling-based methods achieve better stability guarantees**    We introduce the Stability Certification Algorithm (SCA) in Section 3, a simple, model-agnostic, sampling-efficient approach for certifying *both* hard and soft stability with rigorous statistical guarantees. The key idea is to directly estimate the stability rate, which enables certification for both types of stability. We show in Section 5 that SCA gives stronger certificates than smoothing-based methods like MuS.

**Mild smoothing can theoretically improve stability**    Although SCA is model-agnostic, we find that mild MuS-style smoothing can improve the stability rate while preserving model accuracy. Unlike with MuS, this improvement does not require significantly sacrificing accuracy for smoothness. To study this behavior, we use Boolean analytic techniques to give a novel characterization of stability under smoothing in Section 4 and empirically validate our findings in Section 5.

## 2   Background and Overview

Feature attributions are widely used in explainability due to their simplicity and generality, but they are not without drawbacks. In this section, we first give an overview of feature attributions. We then discuss the existing work on hard stability and introduce soft stability.

### 2.1   Feature Attributions as Explanations

Let $f : \mathbb{R}^n \to \mathbb{R}^m$ be a classifier that maps each input $x \in \mathbb{R}^n$ to a vector of $m$ class scores. A feature attribution method assigns an attribution score $\alpha_i \in \mathbb{R}$ to each input feature $x_i$ that indicates its importance to the prediction $f(x)$. The notion of importance is method-dependent: in gradient-based methods [59, 63], $\alpha_i$ typically denotes the gradient at $x_i$, while in Shapley-based methods [39, 62], it represents the Shapley value of $x_i$. For real-valued attribution scores, it is common to convert them into binary vectors by selecting the top-$k$ highest-scoring features [46, 51].

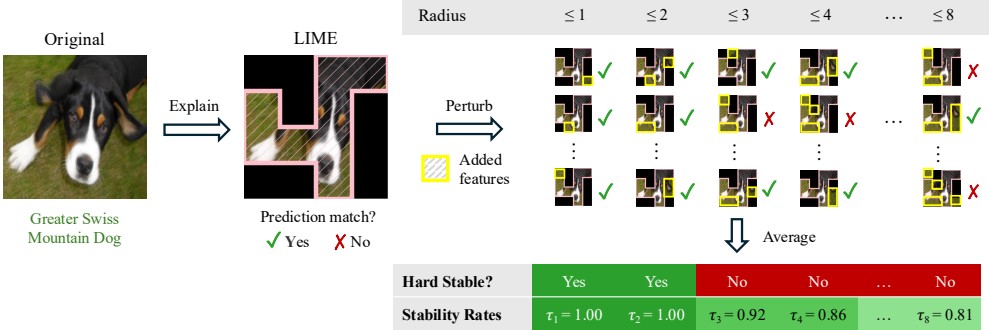

Figure 2: **Soft stability offers a fine-grained measure of robustness.** For Vision Transformer [17], LIME's explanation [51] is only hard stable up to radius $r \leq 2$. In contrast to hard stability's binary decision at each $r$, soft stability uses the *stability rate* $\tau_r$ to quantify the fraction of $\leq r$-sized perturbations that preserve the prediction, yielding a more fine-grained view of explanation stability. Note that hard stability (when $\tau_r = 1$) is a form of adversarial robustness tailored for feature attributions.

## 2.2 Hard Stability and Soft Stability

Many evaluation metrics exist for binary-valued feature attributions [3]. To compare two attributions $\alpha, \alpha' \in \{0, 1\}^n$, it is common to check whether they induce the same prediction with respect to a given classifier $f : \mathbb{R}^n \to \mathbb{R}^m$ and input $x \in \mathbb{R}^n$. Let $(x \odot \alpha) \in \mathbb{R}^n$ be the $\alpha$-masked variant of $x$, where $\odot$ is the coordinate-wise product of two vectors. We write $f(x \odot \alpha) \cong f(x \odot \alpha')$ to mean that the masked inputs $x \odot \alpha$ and $x \odot \alpha'$ yield the same prediction under $f$. This way of evaluating explanations is related to notions of faithfulness, fidelity, and consistency in the explainability literature [47], and is commonly used in both vision [26] and language [40, 71].

It is often desirable that two similar attributions yield the same prediction [72]. While similarity can be defined in various ways, such as overlapping feature sets [47], we focus on additive perturbations. Given an explanation $\alpha$, we define an additive perturbation $\alpha'$ as one that includes more features than $\alpha$. This is based on the intuition that adding information (features) to a high-quality explanation should not significantly affect the classifier's prediction.

**Definition 2.1** (Additive Perturbations). For an attribution $\alpha$ and integer-valued radius $r \geq 0$, define $r$-additive perturbation set of $\alpha$ as:

$$\Delta_r(\alpha) = \{\alpha' \in \{0, 1\}^n : \alpha' \geq \alpha, |\alpha' - \alpha| \leq r\}, \tag{1}$$

where $\alpha' \geq \alpha$ iff each $\alpha'_i \geq \alpha_i$ and $|\cdot|$ counts the non-zeros in a binary vector (i.e., the $\ell^0$ norm).

The binary vectors in $\Delta_r(\alpha)$ represent attributions (explanations) that superset $\alpha$ by at most $r$ features. This lets us study explanation robustness by studying how a more inclusive selection of features affects the classifier's prediction. A natural way to formalize this is through stability: an attribution $\alpha$ is stable with respect to $f$ and $x$ if adding a small number of features does not alter (or rarely alters) the prediction. One such formulation of this idea is *hard stability*.

**Definition 2.2** (Hard Stability [2] [70]). For a classifier $f$ and input $x$, the explanation $\alpha$ is *hard-stable* with radius $r$ if: $f(x \odot \alpha') \cong f(x \odot \alpha)$ for all $\alpha' \in \Delta_r$.

In essence, hard stability is a form of adversarial robustness tailored for feature attributions. The certification process verifies that an explanation $\alpha$ is robust against a worst-case adversary who adds up to $r$ features to make it fail. Specifically, $\alpha$ has a certified hard stability radius of $r$ if one can formally prove that all perturbations $\alpha' \in \Delta_r(\alpha)$ induce the same prediction. While this guarantee is powerful, its certification is not straightforward, as existing algorithms suffer from costly trade-offs that we later discuss in Section 3.1. This practical barrier motivated our development of *soft stability*: a probabilistic relaxation of hard stability that offers a more tractable yet meaningful way to quantify robustness. [3]

---

[2]Xue et al. [70] equivalently call this "incrementally stable" and define "stable" as a stricter property.

[3]Although probabilistic notions of robust explainability have been studied in the literature [12, 52, 66, 67], soft stability stands out as a one-sided notion of robustness.

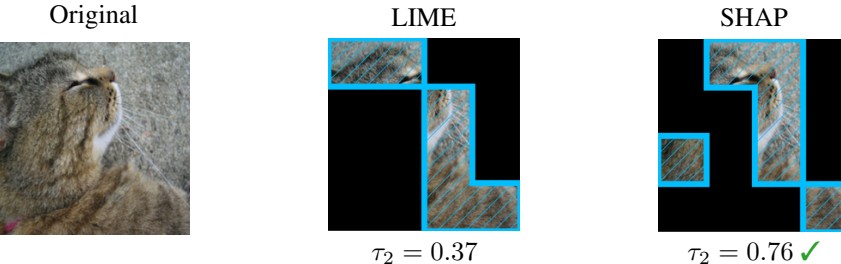

| Original | LIME | SHAP |
|----------|------|------|
| | $\tau_2 = 0.37$ | $\tau_2 = 0.76$ ✓ |

Figure 3: **Similar explanations may have different stability rates.** Despite visual similarities, the explanations generated by LIME [51] (middle) and SHAP [39] (right), both in blue, have different stability rates at radius $r = 2$. In this example, SHAP's explanation is more stable than LIME's.

**Definition 2.3** (Soft Stability). For a classifier $f$ and input $x$, define the *stability rate* $\tau_r(f, x, \alpha)$ as the probability that the prediction remains unchanged when $\alpha$ is perturbed by up to $r$ features:

$$\tau_r(f, x, \alpha) = \Pr_{\alpha' \sim \Delta_r} [f(x \odot \alpha') \cong f(x \odot \alpha)], \quad \text{where } \alpha' \sim \Delta_r \text{ is uniformly sampled.} \quad (2)$$

When $f, x, \alpha$ are clear from the context, we will simply write $\tau_r$ for brevity. An important aspect of soft stability is that it can distinguish between the robustness of two similar explanations. In Figure 3, for example, LIME and SHAP find significantly overlapping explanations that have very different stability rates. We further study the stability rate of different explanation methods in Section 5.

**Relation Between Hard and Soft Stability** Soft stability is a probabilistic relaxation of hard stability, with $\tau_r = 1$ recovering the hard stability condition. Conversely, hard stability is a valid but coarse lower bound on the stability rate: if $\tau_r < 1$, then the explanation is not hard stable at radius $r$. This relation implies that any certification for one kind of stability can be adapted for the other.

## 3 Certifying Stability: Challenges and Algorithms

We begin by discussing the limitations of existing hard stability certification methods, particularly those based on smoothing, such as MuS [70]. We then introduce the Stability Certification Algorithm (SCA) in Equation (3), providing a simple, model-agnostic, and sample-efficient way to certify both hard (Theorem 3.2) and soft (Theorem 3.1) stability at all perturbation radii.

### 3.1 Limitations in (MuS) Smoothing-based Hard Stability Certification

Existing hard stability certifications rely on a classifier's *Lipschitz constant*, which is a measure of sensitivity to input perturbations. While the Lipschitz constant is useful for robustness analysis [14], it is often intractable to compute and difficult to approximate [20, 43, 64, 69]. To address this, Xue et al. [70] construct smoothed classifiers with analytically known Lipschitz constants. Given a classifier $f$, its smoothed variant $\tilde{f}$ is defined as the average prediction over perturbed inputs: $\tilde{f}(x) = \frac{1}{N} \sum_{i=1}^{N} f(x^{(i)})$, where $x^{(1)}, \ldots, x^{(N)} \sim \mathcal{D}(x)$ are perturbations of $x$. If $\mathcal{D}$ is appropriately chosen, then the smoothed classifier $\tilde{f}$ has a known Lipschitz constant $\kappa$ that allows for efficient certification. We review MuS smoothing in Definition 4.1 and its hard stability certificates in Theorem C.1.

**Smoothing has severe performance trade-offs** A key limitation of smoothing-based certificates is that the stability guarantees apply to $\tilde{f}$ rather than $f$. Typically, the smoother the classifier, the stronger its guarantees (larger certified radii), but this comes at the cost of accuracy. This is because smoothing reduces a classifier's sensitivity, making it harder to distinguish between classes [6, 25].

**Smoothing-based hard stability is conservative** Even when a smoothing-based certified radius is obtained, it is often conservative. The main reason is that this approach depends on a global property, the Lipschitz constant $\kappa$, to make guarantees about local perturbations $\alpha' \sim \Delta_r(\alpha)$. In particular, the certified hard stability radius of $\tilde{f}$ scales as $\mathcal{O}(1/\kappa)$, which we elaborate on in Theorem C.1.

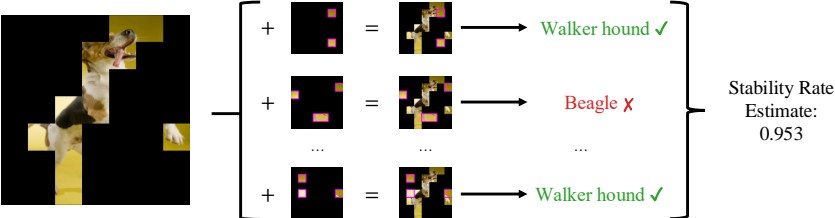

Figure 4: **The stability certification algorithm (SCA).** Given an explanation $\alpha \in \{0,1\}^n$ for a classifier $f$ and input $x \in \mathbb{R}^n$, we estimate the stability rate $\tau_r$ as follows. First, sample perturbed masks $\alpha' \sim \Delta_r(\alpha)$ uniformly with replacement. Then, compute the empirical stability rate $\hat{\tau}_r$, defined as the fraction of samples that preserve the prediction: $\hat{\tau}_r = \frac{1}{N} \sum_{\alpha'} \mathbf{1}[f(x \odot \alpha') \cong f(x \odot \alpha)]$. With a properly chosen sample size $N$, both hard and soft stability can be certified with statistical guarantees.

## 3.2 Sampling-based Algorithms for Certifying Stability

Our key insight is that both soft and hard stability can be certified by directly estimating the stability rate through sampling. This leads to a simple algorithm, illustrated in Figure 4 and formalized below:

$$\hat{\tau}_r = \frac{1}{N} \sum_{i=1}^{N} \mathbf{1}\left[f(x \odot \alpha^{(i)}) \cong f(x \odot \alpha)\right], \quad \text{where } \alpha^{(1)}, \dots, \alpha^{(N)} \sim \Delta_r(\alpha) \text{ are sampled i.i.d. (3)}$$

The estimator $\hat{\tau}_r$ provides a statistical approximation of soft stability. With an appropriate sample size $N$, this estimate yields formal guarantees for both hard and soft stability.

**Theorem 3.1** (Certifying Soft Stability with SCA). *Let $\hat{\tau}_r$ be the stability rate estimator defined in (3), computed with $N \geq \frac{\log(2/\delta)}{2\varepsilon^2}$ for any confidence parameter $\delta > 0$ and error tolerance $\varepsilon > 0$. Then, with probability at least $1 - \delta$, the estimator satisfies $|\hat{\tau}_r - \tau_r| \leq \varepsilon$.*

*Proof.* The result follows by applying Hoeffding's inequality to the empirical mean of independent Bernoulli random variables $X^{(1)}, \dots, X^{(N)}$, where each $X^{(i)} = \mathbf{1}[f(x \odot \alpha^{(i)}) \cong f(x \odot \alpha)]$. $\square$

SCA can also certify hard stability by noting that $\hat{\tau}_r = 1$ implies a high-confidence guarantee.

**Theorem 3.2** (Certifying Hard Stability with SCA). *Let $\hat{\tau}_r$ be the stability rate estimator defined in Equation (3), computed with sample size $N \geq \frac{\log(\delta)}{\log(1-\varepsilon)}$ for any confidence parameter $\delta > 0$ and error tolerance $\varepsilon > 0$. If $\hat{\tau}_r = 1$, then with probability at least $1 - \delta$, a uniformly sampled $\alpha' \sim \Delta_r(\alpha)$ violates hard stability with probability at most $\varepsilon$.*

*Proof.* We bound the probability of the worst-case event, where the explanation is not hard stable at radius $r$, meaning $\tau_r < 1 - \varepsilon$, yet the estimator satisfies $\hat{\tau}_r = 1$. Because each $\alpha^{(i)} \sim \Delta_r$ is uniformly sampled, this event occurs with probability

$$\Pr\left[\hat{\tau}_r = 1 \,|\, \tau_r < 1 - \varepsilon\right] \leq (1 - \varepsilon)^N \leq \delta,$$

which holds whenever $N \geq \log(\delta)/\log(1 - \varepsilon)$. $\square$

In both hard and soft stability certification, the required sample size $N$ depends only on $\varepsilon$ and $\delta$, as $\tau_r$ is a one-dimensional statistic. Notably, certifying hard stability requires fewer samples, since the event being verified is simpler. In both settings, SCA provides a simple alternative to MuS that does not require smoothing.

**Implementing SCA** The main computational challenge is in sampling $\alpha' \sim \Delta_r(\alpha)$ uniformly. When $r \leq n - |\alpha|$, this may be done by: (1) sampling a perturbation size $k \sim \{0, 1, \dots, r\}$ with probability $\binom{n-|\alpha|}{k}/|\Delta_r(\alpha)|$, where $|\Delta_r(\alpha)| = \sum_{i=0}^{r} \binom{n-|\alpha|}{i}$; and then (2) uniformly selecting $k$ zero positions in $\alpha$ to flip to one. To avoid numerical instability from large binomial coefficients, we use a Gumbel softmax reparametrization [27] to sample in the log probability space.

# 4 Theoretical Link Between Stability and Smoothing

While SCA does *not* require smoothing to certify stability, we find that applying mild MuS-style smoothing can improve the stability rate while incurring only a minor accuracy trade-off. While this improvement is unsurprising, it is notable that the underlying smoothing mechanism is *discrete*. In contrast, most prior work relies on *continuous* noise distributions [14]. Below, we introduce this discrete smoothing method, MuS, wherein the main idea is to promote robustness to feature inclusion and exclusion by averaging predictions over randomly masked (dropped) inputs.

**Definition 4.1** (MuS[4] (Random Masking)). For any classifier $f$ and smoothing parameter $\lambda \in [0, 1]$, define the random masking operator $M_\lambda$ as:

$$M_\lambda f(x) = \mathop{\mathbb{E}}_{z \sim \mathrm{Bern}(\lambda)^n} f(x \odot z), \quad \text{where } z_1, \ldots, z_n \sim \mathrm{Bern}(\lambda) \text{ are i.i.d. samples.} \tag{4}$$

Here, $\tilde{f} = M_\lambda f$ is the smoothed classifier, where each feature is kept with probability $\lambda$. A smaller $\lambda$ implies stronger smoothing: at $\lambda = 1$, we have $\tilde{f} = f$; at $\lambda = 1/2$, half the features of $x \odot z$ are dropped on average; at $\lambda = 0$, $\tilde{f}$ reduces to a constant classifier. We summarize our main results in Section 4.1 with details in Section 4.2, and extended discussions in Appendix A and Appendix B.

## 4.1 Summary of Theoretical Results

Our main theoretical tooling is Boolean function analysis [48], which studies real-valued functions of Boolean-valued inputs. To connect this with evaluating explanations: for any classifier $f : \mathbb{R}^n \to \mathbb{R}^m$ and input $x \in \mathbb{R}^n$, define the masked evaluation $f_x(\alpha) = f(x \odot \alpha)$. Such $f_x : \{0, 1\}^n \to \mathbb{R}^m$ is then a Boolean function, for which the random masking (MuS) operator $M_\lambda$ is well-defined because $M_\lambda f(x \odot \alpha) = M_\lambda f_x(\alpha)$. To simplify our analysis, we consider a simpler form of prediction agreement for classifiers of the form $f_x : \{0, 1\}^n \to \mathbb{R}$, where for $\alpha' \sim \Delta_r(\alpha)$ let:

$$f_x(\alpha') \cong f_x(\alpha) \quad \text{if} \quad |f_x(\alpha') - f_x(\alpha)| \leq \gamma, \tag{5}$$

where $\gamma$ is the distance to the decision boundary. [5] This setup can be derived from a general $m$-class classifier once the $x$ and $\alpha$ are given. In summary, we establish the following.

**Theorem 4.2** (Smoothed Stability, Informal of Theorem B.4). *Smoothing improves the lower bound on the stability rate by shrinking its gap to $1$ by a factor of $\lambda$. Consider any classifier $f_x$ and attribution $\alpha$ that satisfy Equation (5), and let $Q$ depend on the monotone weights of $f_x$, then:*

$$1 - \frac{Q}{\gamma} \leq \tau_r(f_x, \alpha) \implies 1 - \frac{\lambda Q}{\gamma} \leq \tau_r(M_\lambda f_x, \alpha). \tag{6}$$

Theoretically, smoothing improves the worst-case stability rate by a factor of $\lambda$. Empirically, we observe that smoothed classifiers tend to be more stable. Interestingly, we found it challenging to bound the stability rate of $M_\lambda$-smoothed classifiers using standard Boolean analytic techniques, such as those in widely used references like [48]. This motivated us to develop novel analytic tooling to study stability under smoothing, which we discuss next.

## 4.2 Challenges with Standard Boolean Analytic Tooling and New Techniques

It is standard to study Boolean functions via their Fourier expansion. For any $h : \{0, 1\}^n \to \mathbb{R}$, its Fourier expansion exists uniquely as a linear combination over the subsets of $[n] = \{1, \ldots, n\}$:

$$h(\alpha) = \sum_{S \subseteq [n]} \widehat{h}(S) \chi_S(\alpha), \tag{7}$$

where each $\chi_S(\alpha)$ is a Fourier basis function with weight $\widehat{h}(S)$, respectively defined as:

$$\chi_S(\alpha) = \prod_{i \in S} (-1)^{\alpha_i}, \quad \chi_\emptyset(\alpha) = 1, \quad \widehat{h}(S) = \frac{1}{2^n} \sum_{\alpha \in \{0,1\}^n} h(\alpha) \chi_S(\alpha). \tag{8}$$

---

[4]We use the terms *MuS*, *random masking*, *smoothing*, and $M_\lambda$ interchangeably, depending on the context.

[5]In the special case where the model outputs a sorted probability vector with $p_1 \geq p_2 \geq \cdots \geq p_m$, we let $\gamma = (p_1 - p_2)/2$. This is half the gap between the top two classes, which ensures that even if $p_1$ decreases by $\gamma$, it remains the highest class.

The Fourier expansion makes all the $k = 0, 1, \ldots, n$ degree (order) interactions between input bits explicit. For example, the AND function $h(\alpha_1, \alpha_2) = \alpha_1 \wedge \alpha_2$ is uniquely expressible as:

$$h(\alpha_1, \alpha_2) = \frac{1}{4}\chi_\emptyset(\alpha) - \frac{1}{4}\chi_{\{1\}}(\alpha) - \frac{1}{4}\chi_{\{2\}}(\alpha) + \frac{1}{4}\chi_{\{1,2\}}(\alpha). \tag{9}$$

To study how linear operators act on Boolean functions, it is common to isolate their effect on each term. With respect to the standard Fourier basis, the operator $M_\lambda$ acts as follows.

**Theorem 4.3.** *For any standard basis function $\chi_S$ and smoothing parameter $\lambda \in [0, 1]$,*

$$M_\lambda \chi_S(\alpha) = \sum_{T \subseteq S} \lambda^{|T|}(1 - \lambda)^{|S-T|}\chi_T(\alpha). \tag{10}$$

*For any function $h : \{0, 1\}^n \to \mathbb{R}$, its smoothed variant $M_\lambda h$ has the Fourier expansion*

$$M_\lambda h(\alpha) = \sum_{T \subseteq [n]} \widehat{M_\lambda h}(T)\chi_T(\alpha), \quad \text{where} \quad \widehat{M_\lambda h}(T) = \lambda^{|T|} \sum_{S \supseteq T} (1 - \lambda)^{|S-T|}\widehat{h}(S). \tag{11}$$

This result shows that smoothing redistributes weights from each term $S$ down to all of its subsets $T \subseteq S$, scaled by a binomial decay $\mathsf{Bin}(|S|, \lambda)$. However, this behavior introduces significant complexity in the algebraic manipulations and is distinct from that of other operators commonly studied in literature, making it difficult to analyze stability with existing techniques.

Although one could, in principle, study stability using the standard basis, we found that the *monotone basis* was better suited to describing the inclusion and exclusion of features. While this basis is also known in game theory as *unanimity functions*, its use in analyzing stability and smoothing is novel.

**Definition 4.4** (Monotone Basis). For each subset $T \subseteq [n]$, define its monotone basis function as:

$$\mathbf{1}_T(\alpha) = \begin{cases} 1 & \text{if } \alpha_i = 1 \text{ for all } i \in T \text{ (all features of } T \text{ are present)}, \\ 0 & \text{otherwise (any feature of } T \text{ is absent).} \end{cases} \tag{12}$$

The monotone basis provides a direct encoding of set inclusion, where the example of conjunction is now concisely represented as $\mathbf{1}_{\{1,2\}}(\alpha_1, \alpha_2) = \alpha_1 \wedge \alpha_2$. Similar to the standard basis, the monotone basis also admits a unique *monotone expansion* for any function $h : \{0, 1\}^n \to \mathbb{R}$ and takes the form:

$$h(\alpha) = \sum_{T \subseteq [n]} \widetilde{h}(T)\mathbf{1}_T(\alpha), \quad \text{where} \quad \widetilde{h}(T) = h(T) - \sum_{S \subsetneq T} \widetilde{h}(S), \quad \widetilde{h}(\emptyset) = h(\mathbf{0}_n), \tag{13}$$

where $\widetilde{h}(T)$ are the recursively defined monotone weights at each $T \subseteq [n]$, with $h(T)$ being the evaluation of $h$ on the natural $\{0, 1\}^n$-valued representation of $T$. A key property of the monotone basis is that the action of $M_\lambda$ is now a point-wise contraction at each $T$.

**Theorem 4.5.** *For any function $h : \{0, 1\}^n \to \mathbb{R}$, subset $T \subseteq [n]$, and $\lambda \in [0, 1]$, the smoothed classifier experiences a spectral contraction of*

$$\widetilde{M_\lambda h}(T) = \lambda^{|T|}\widetilde{h}(T), \tag{14}$$

*where $\widetilde{M_\lambda h}(T)$ and $\widetilde{h}(T)$ are the monotone basis coefficients of $M_\lambda h$ and $h$ at subset $T$, respectively.*

In contrast to smoothing in the standard basis (Theorem 4.3), smoothing in the monotone basis exponentially decays each weight by a factor of $\lambda^{|T|}$, which better aligns with the motifs of existing techniques. [6] As previewed in Theorem 4.2, the stability rate of smoothed classifiers can be bounded via the monotone weights of degree $\leq r$, which we further discuss in Appendix B.

## 5 Experiments

We evaluate the advantages of SCA over MuS, which is currently the only other stability certification algorithm. We also study how stability guarantees vary across vision and language tasks, as well as across explanation methods. Moreover, we show that mild smoothing, defined as $\lambda \geq 0.5$ for Definition 4.1, often improves stability while preserving accuracy. We summarize our key findings here and defer full technical details and additional experiments to Appendix C.

---

[6]The standard smoothing operator is random flipping: let $T_\rho h(\alpha) = \mathbb{E}_{z \sim \mathsf{Bern}(q)^n}[h((\alpha + z) \bmod 2)]$ for any $\rho \in [0, 1]$ and $q = (1 - \rho)/2$. Then, the standard Fourier basis contracts as $T_\rho \chi_S(\alpha) = \rho^{|S|}\chi_S(\alpha)$.

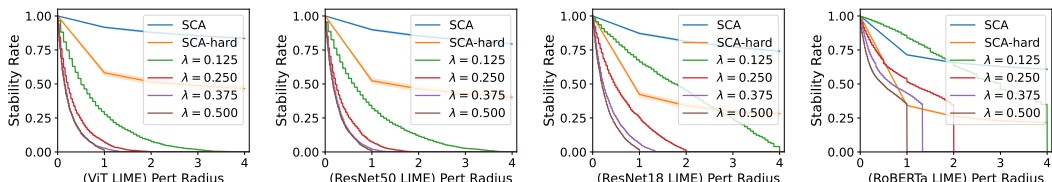

Figure 5: **SCA certifies more than MuS.** Soft stability certificates obtained through SCA are stronger than those obtained from MuS, which quickly become vacuous as the perturbation size grows. When using MuS with smoothing parameter $\lambda$, guarantees only exist for perturbation radii $\leq 1/2\lambda$. Moreover, the smaller the $\lambda$, the worse the smoothed classifier accuracy, see Figure 8.

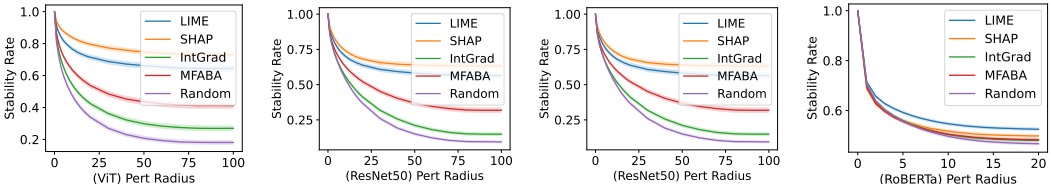

Figure 6: **Soft stability varies across explanation methods.** For vision models, LIME and SHAP yield higher stability rates than gradient-based methods, with all methods outperforming the random baseline. On RoBERTa, however, the methods are less distinguishable. Note that a perturbation of size 100 affects over half the features in a patched image input with $n = 196$ features.

**Experimental Setup** We used Vision Transformer (ViT) [17] and ResNet50/18 [24] as our vision models and RoBERTa [38] as our language model. For datasets, we used a 2000-image subset of ImageNet (2 images per class) and six subsets of TweetEval (emoji, emotion, hate, irony, offensive, sentiment), totaling 10653 samples. Images of size $3 \times 224 \times 224$ were segmented into $16 \times 16$ patches, for $n = 196$ features per image. For text, each token was treated as one feature. We used five feature attribution methods: LIME [51], SHAP [39], Integrated Gradients [63], MFABA [75], and a random baseline. We selected the top-25% of features as the explanation.

**Question 1: How do SCA's guarantees compare to those from MuS?** We begin by comparing the SCA-based stability guarantees to those from MuS. To facilitate comparison, we derive stability rates for MuS-based hard stability certificates (Theorem C.1) using the following formulation:

$$\text{Stability rate at radius } r = \frac{|\{(x,\alpha) : \text{CertifiedRadius}(M_\lambda f_x, \alpha) \geq r\}|}{\text{Total number of } x\text{'s}}. \quad (15)$$

In Figure 5, we present results for LIME across different MuS smoothing parameters $\lambda$, along with the SCA-based soft (Theorem 3.1) and hard (Theorem 3.2) stability certificates. SCA yields non-trivial guarantees even at larger perturbation radii, whereas MuS-based certificates become vacuous beyond a radius of $1/2\lambda$. A smaller $\lambda$ improves MuS guarantees but significantly degrades accuracy (see Figure 8), resulting in certificates for less accurate classifiers. Section Appendix C.2 presents an extended comparison of SCA and MuS over various explanations, where we observe similar trends.

**Question 2: How does stability vary across explanation methods?** We next show in Figure 6 how the SCA-certified stability rate varies across different explanation methods. Soft stability can effectively distinguish between explanation methods in vision, with LIME and SHAP yielding the highest stability rates. However, this distinction is less clear for RoBERTa and for MuS-based hard stability certificates, further studied in Appendix C.3. Furthermore, we show ablations on the top-$k$ feature selection in Appendix C.4.

**Question 3: How well does mild smoothing ($\lambda \geq 0.5$) improve stability?** We next empirically study the relation between stability and mild smoothing, for which $\lambda \geq 0.5$ is too large to obtain hard stability certificates. We show in Figure 7 the stability rate at different $\lambda$, where we used 32 Bernoulli samples to compute smoothing (Definition 4.1). We used 200 samples from our subset of ImageNet and 200 samples from TweetEval that had at least 40 tokens, and a random attribution to select 25%

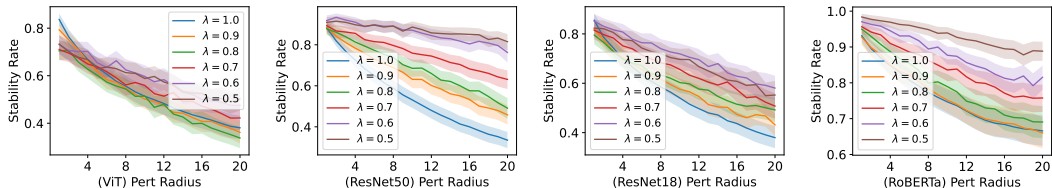

Figure 7: **Mild smoothing ($\lambda \geq 0.5$) can improve stability.** For vision, this is most prominent for ResNet50 and ResNet18. While transformers also benefit, RoBERTa improves more than ViT.

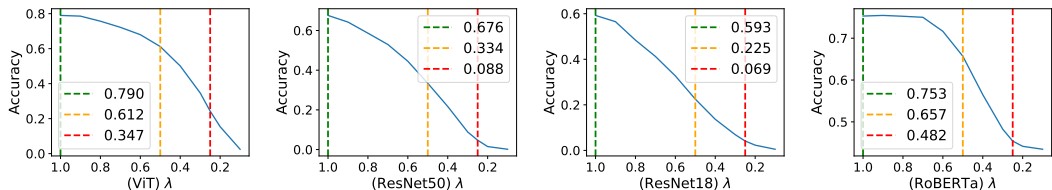

Figure 8: **Mild smoothing ($\lambda \geq 0.5$) preserves accuracy.** We report accuracy at three key smoothing levels: ($\lambda = 1.0$, in green) the original, unsmoothed classifier; ($\lambda = 0.5$, in orange) a mildly smoothed classifier, the largest $\lambda$ for which hard stability certificates can be obtained; ($\lambda = 0.25$, in red) a heavily smoothed classifier, where MuS can only certify at most a perturbation radius of size 2.

of the features. We see that smoothing generally improves stability, and we study setups with larger perturbation radii Appendix C.5.

**Question 4: How well do mildly smoothed classifiers trade off accuracy?** We analyze the impact of MuS smoothing on classifier accuracy in Figure 8 and highlight three key values: the original, unmodified classifier accuracy ($\lambda = 1.0$), the largest smoothing parameter usable in the certification of hard stability ($\lambda = 0.5$), and the smoothing parameter used in many hard stability experiments of [70] ($\lambda = 0.25$). We used 64 Bernoulli samples to compute smoothing (Definition 4.1). These results demonstrate the utility of mild smoothing. In particular, transformers (ViT, RoBERTa) exhibit a more gradual decline in accuracy, likely because their training involves random masking.

## 6 Related Work

**Feature-based Explanations** Feature attributions are a popular class of explanation methods. Early examples include gradient saliency [59], LIME [51], SHAP [39], Integrated Gradients [63], and SmoothGrad [61]. More recent works include DIME [42], LAFA [73], CAFE [15], DoRaR [50], MFABA [75], various Shapley value-based methods [62], and methods based on influence functions [10, 33]. While feature attributions are commonly associated with vision models, they are also used in language [41] and time series modeling [56]. However, they have known limitations [11, 18, 44, 60]. We refer to [45, 47, 58] for general surveys, to [32, 49] for surveys on explainability in medicine, and to [4, 53] for surveys on explainability in law.

**Evaluating and Certifying Explanations** There is much work on empirically evaluating feature attributions [1–3, 16, 28, 31, 47, 54, 74], with various notions of robustness [21, 29]. Probabilistic notions of robust explainability are explored in [12, 52, 66, 67], though stability is notable in that it is a form of one-sided robustness. There is also growing interest in certified explanations. For instance, certifying that an explanation is robust to adding [70] and removing [36] features, that it is minimal [9, 12], or that the attribution scores are robustly ranked [22]. A related notion of probabilistic guarantees exists for analyzing the explanation method itself [30], which quantifies how much the feature attribution changes as the input is perturbed. However, the literature on certified explanations is still emergent.

## 7 Discussion

Many perturbations relevant to explainability are inherently discrete, such as feature removal or token substitution. This contrasts with continuous perturbations, e.g., Gaussian noise, which are more commonly studied in adversarial robustness literature. This motivates the development of new techniques for discrete robustness, such as those inspired by Boolean analysis. In our case, this approach enabled us to shift away from traditional Lipschitz-based techniques to provide an alternative analysis of robustness. Our work highlights the potential of discrete methods in explainability.

Our stability framework generalizes adversarial robustness. Hard stability, the case where $\tau_r = 1$, is certified robustness against an adversary adding up to $r$ features. However, this discrete, structural attack model differs from the continuous $\ell_p$-norm perturbations common in adversarial robustness. Soft stability offers a more nuanced evaluation, where the stability rate $\tau_r$ quantifies an explanation's success under random additive attacks, providing a richer characterization of its robustness. Thus, in this view, stability itself can be interpreted as a form of adversarial robustness. If an explanation achieves a stability rate of 1 at some radius, it is adversarially robust up to that perturbation radius. Consequently, high stability rates, ideally at 1, are desirable indicators of robust explanations.

## 8 Conclusion

Soft stability is a form of stability that enables fine-grained measures of explanation robustness to additive perturbations. We introduce SCA to certify stability and show that it yields stronger guarantees than existing smoothing-based certifications, such as MuS. Although SCA does not require smoothing, mild smoothing can improve stability at little cost to accuracy, and we use Boolean analytic tooling to explain this phenomenon. We validate our findings with experiments on vision and language models across a range of explanation methods.

Potential directions include adaptive smoothing based on feature importance and ranking [22], as well as selectively smoothing only parts of the features [57]. One could also study stability-regularized training in relation to adversarial training. Other directions include robust explanations through other families of probabilistic guarantees, such as those based on conformal prediction [5, 8, 13, 35]. Additionally, it would be interesting to investigate how well explanation stability aligns with human evaluations of quality.

**Acknowledgements**   This research was partially supported by the ARPA-H program on Safe and Explainable AI under the grant D24AC00253-00, by NSF award CCF 2313010, by the AI2050 program at Schmidt Sciences, by an Amazon Research Award Fall 2023, by an OpenAI SuperAlignment grant, and Defense Advanced Research Projects Agency's (DARPA) SciFy program (Agreement No. HR00112520300). The views expressed are those of the author and do not reflect the official policy or position of the Department of Defense or the U.S. Government.

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

# A  Analysis of Smoothing with Standard Techniques

In this appendix, we analyze the smoothing operator $M_\lambda$ using classical tools from Boolean function analysis. Specifically, we study how smoothing redistributes the spectral mass of a function by examining its action on standard Fourier basis functions. This sets up the foundation for our later motivation to introduce a more natural basis in Appendix B. First, recall the definition of the random masking-based smoothing operator.

**Definition A.1** (MuS [70] (Random Masking)). *For any classifier $f : \mathbb{R}^n \to \mathbb{R}^m$ and smoothing parameter $\lambda \in [0, 1]$, define the random masking operator $M_\lambda$ as:*

$$M_\lambda f(x) = \underset{z \sim \text{Bern}(\lambda)^n}{\mathbb{E}} f(x \odot z), \quad \text{where } z_1, \ldots, z_n \sim \text{Bern}(\lambda) \text{ are i.i.d. samples.} \tag{16}$$

To study $M_\lambda$ via Boolean function analysis, we fix the input $x \in \mathbb{R}^n$ and view the masked classifier $f_x(\alpha) = f(x \odot \alpha)$ as a Boolean function $f_x : \{0, 1\}^n \to \mathbb{R}^m$. In particular, we have the following:

$$M_\lambda f(x \odot \alpha) = M_\lambda f_x(\alpha) = M_\lambda f_{x \odot \alpha}(\mathbf{1}_n). \tag{17}$$

This relation is useful from an explainability perspective because it means that features not selected by $\alpha$ (the $x_i$ where $\alpha_i = 0$) will not be seen by the classifier. In other words, this prevents a form of information leakage when evaluating the informativeness of a feature selection.

## A.1  Background on Boolean Function Analysis

A key approach in Boolean function analysis is to study functions of the form $h : \{0, 1\}^n \to \mathbb{R}$ by their unique *Fourier expansion*. This is a linear combination indexed by the subsets $S \subseteq [n]$ of form:

$$h(\alpha) = \sum_{S \subseteq [n]} \widehat{h}(S) \chi_S(\alpha), \tag{18}$$

where each $\chi_S(\alpha)$ is a Fourier basis function, also called the standard basis function, with weight $\widehat{h}(S)$. These quantities are respectively defined as:

$$\chi_S(\alpha) = \prod_{i \in S} (-1)^{\alpha_i}, \quad \chi_\emptyset(\alpha) = 1, \quad \widehat{h}(S) = \frac{1}{2^n} \sum_{\alpha \in \{0,1\}^n} h(\alpha) \chi_S(\alpha). \tag{19}$$

The functions $\chi_S : \{0, 1\}^n \to \{\pm 1\}$ form an orthonormal basis on $\{0, 1\}^n$ in the sense that:

$$\langle \chi_S, \chi_T \rangle = \underset{\alpha \sim \text{Bern}(1/2)^n}{\mathbb{E}} [\chi_S(\alpha) \chi_T(\alpha)] = \frac{1}{2^n} \sum_{\alpha \in \{0,1\}^n} \chi_S(\alpha) \chi_T(\alpha) = \begin{cases} 1 & \text{if } S = T, \\ 0 & \text{if } S \neq T. \end{cases} \tag{20}$$

Consequently, all of the $2^n$ weights $\widehat{h}(S)$ (one for each $S \subseteq [n]$) are uniquely determined by the $2^n$ values of $h(\alpha)$ (one for each $\alpha \in \{0, 1\}^n$) under the linear relation $\widehat{h}(S) = \langle h, \chi_S \rangle$ as in Equation (19). For example, one can check that the function $h(\alpha_1, \alpha_2) = \alpha_1 \wedge \alpha_2$ is uniquely expressible in this basis as:

$$h(\alpha_1, \alpha_2) = \frac{1}{4} \chi_\emptyset(\alpha) - \frac{1}{4} \chi_{\{1\}}(\alpha) - \frac{1}{4} \chi_{\{2\}}(\alpha) + \frac{1}{4} \chi_{\{1,2\}}(\alpha). \tag{21}$$

We defer to O'Donnell [48] for a more comprehensive introduction to Boolean function analysis.

## A.2  Basic Results in the Standard Basis

We now study how smoothing affects stability by analyzing how $M_\lambda$ transforms Boolean functions in the standard Fourier basis. A common approach is to examine how $M_\lambda$ acts on each basis function $\chi_S$, and we show that smoothing causes a spectral mass shift from higher-order to lower-order terms.

**Lemma A.2.** *For any standard basis function $\chi_S$ and $\lambda \in [0, 1]$,*

$$M_\lambda \chi_S(\alpha) = \sum_{T \subseteq S} \lambda^{|T|} (1 - \lambda)^{|S - T|} \chi_T(\alpha). \tag{22}$$

*Proof.* We first expand the definition of $\chi_S(\alpha)$ to derive:

$$M_\lambda \chi_S(\alpha) = \mathop{\mathbb{E}}_z \prod_{i \in S} (-1)^{\alpha_i z_i} \tag{23}$$

$$= \prod_{i \in S} \mathop{\mathbb{E}}_z (-1)^{\alpha_i z_i} \qquad \text{(by independence of } z_1, \ldots, z_n) $$

$$= \prod_{i \in S} [(1 - \lambda) + \lambda(-1)^{\alpha_i}], \tag{24}$$

We then use the distributive property (i.e., expanding products over sums) to rewrite the product $\prod_{i \in S}(\cdots)$ as a summation over $T \subseteq S$ to get

$$M_\lambda \chi_S(\alpha) = \sum_{T \subseteq S} \left( \prod_{j \in S-T} (1 - \lambda) \right) \left( \prod_{i \in T} \lambda(-1)^{\alpha_i} \right) \tag{25}$$

$$= \sum_{T \subseteq S} (1 - \lambda)^{|S-T|} \lambda^{|T|} \chi_T(\alpha), \tag{26}$$

where $T$ acts like an enumeration over $\{0, 1\}^n$ and recall that $\chi_T(\alpha) = \prod_{i \in T}(\alpha)$. $\qquad\square$

In other words, $M_\lambda$ redistributes the Fourier weight at each basis $\chi_S$ over to the $2^{|S|}$ subsets $T \subseteq S$ according to a binomial distribution $\mathrm{Bin}(|S|, \lambda)$. Since this redistribution is linear in the input, we can visualize $M_\lambda$ as a $\mathbb{R}^{2^n \times 2^n}$ upper-triangular matrix whose entries are indexed by $T, S \subseteq [n]$, where

$$(M_\lambda)_{T,S} = \begin{cases} \lambda^{|T|}(1 - \lambda)^{|S-T|} & \text{if } T \subseteq S, \\ 0 & \text{otherwise.} \end{cases} \tag{27}$$

Using the example of $h(\alpha_1, \alpha_2) = \alpha_1 \wedge \alpha_2$, the Fourier coefficients of $M_\lambda h$ may be written as:

$$\begin{bmatrix} \widehat{M_\lambda h}(\emptyset) \\ \widehat{M_\lambda h}(\{1\}) \\ \widehat{M_\lambda h}(\{2\}) \\ \widehat{M_\lambda h}(\{1,2\}) \end{bmatrix} = \begin{bmatrix} 1 & (1-\lambda) & (1-\lambda) & (1-\lambda)^2 \\ & \lambda & & \lambda(1-\lambda) \\ & & \lambda & \lambda(1-\lambda) \\ & & & \lambda^2 \end{bmatrix} \begin{bmatrix} \widehat{h}(\emptyset) \\ \widehat{h}(\{1\}) \\ \widehat{h}(\{2\}) \\ \widehat{h}(\{1,2\}) \end{bmatrix} = \frac{1}{4} \begin{bmatrix} (2-\lambda)^2 \\ -\lambda(2-\lambda) \\ -\lambda(2-\lambda) \\ \lambda^2 \end{bmatrix} \tag{28}$$

where recall that $\widehat{h}(S) = 1/4$ for all $S \subseteq \{1, 2\}$. For visualization, it is useful to sort the rows and columns of $M_\lambda$ by inclusion and partition them by degree. Below is an illustrative expansion of $M_\lambda \in \mathbb{R}^{8 \times 8}$ for $n = 3$, sorted by inclusion and partitioned by degree:

|  | $\emptyset$ | $\{1\}$ | $\{2\}$ | $\{3\}$ | $\{1,2\}$ | $\{1,3\}$ | $\{2,3\}$ | $\{1,2,3\}$ |
|---|---|---|---|---|---|---|---|---|
| $\emptyset$ | 1 | $(1-\lambda)$ | $(1-\lambda)$ | $(1-\lambda)$ | $(1-\lambda)^2$ | $(1-\lambda)^2$ | $(1-\lambda)^2$ | $(1-\lambda)^3$ |
| $\{1\}$ | | $\lambda$ | | | $\lambda(1-\lambda)$ | $\lambda(1-\lambda)$ | | $\lambda(1-\lambda)^2$ |
| $\{2\}$ | | | $\lambda$ | | $\lambda(1-\lambda)$ | | $\lambda(1-\lambda)$ | $\lambda(1-\lambda)^2$ |
| $\{3\}$ | | | | $\lambda$ | | $\lambda(1-\lambda)$ | $\lambda(1-\lambda)$ | $\lambda(1-\lambda)^2$ |
| $\{1,2\}$ | | | | | $\lambda^2$ | | | $\lambda^2(1-\lambda)$ |
| $\{1,3\}$ | | | | | | $\lambda^2$ | | $\lambda^2(1-\lambda)$ |
| $\{2,3\}$ | | | | | | | $\lambda^2$ | $\lambda^2(1-\lambda)$ |
| $\{1,2,3\}$ | | | | | | | | $\lambda^3$ |

$$\tag{29}$$

Because the columns of $M_\lambda$ sum to 1, we have the identity:

$$\sum_{T \subseteq [n]} \widehat{M_\lambda h}(T) = \sum_{S \subseteq [n]} \widehat{h}(S), \quad \text{for any function } h : \{0, 1\}^n \to \mathbb{R}. \tag{30}$$

Moreover, $M_\lambda$ may be interpreted as a downshift operator in the sense that: for each $T \subseteq [n]$, the Fourier coefficient $\widehat{M_\lambda h}(T)$ depends only on those of $\widehat{h}(S)$ for $S \supseteq T$. The following result gives a more precise characterization of each $\widehat{M_\lambda h}(T)$ in the standard basis.

**Lemma A.3.** *For any function* $h : \{0,1\}^n \to \mathbb{R}$ *and* $\lambda \in [0,1]$,

$$M_\lambda h(\alpha) = \sum_{T \subseteq [n]} \widehat{M_\lambda h}(T) \chi_T(\alpha), \quad \text{where } \widehat{M_\lambda h}(T) = \lambda^{|T|} \sum_{S \supseteq T} (1-\lambda)^{|S-T|} \widehat{h}(S). \tag{31}$$

*Proof.* This follows by analyzing the $T$-th row of $M_\lambda$ as in Equation (29). Specifically, we have:

$$M_\lambda h(\alpha) = \sum_{S \subseteq [n]} \widehat{h}(S) M_\lambda \chi_S(\alpha) \tag{32}$$

$$= \sum_{S \subseteq [n]} \widehat{h}(S) \sum_{T \subseteq S} \lambda^{|T|} (1-\lambda)^{|S-T|} \chi_T(\alpha) \tag{Lemma A.2}$$

$$= \sum_{T \subseteq [n]} \chi_T(\alpha) \underbrace{\sum_{S \supseteq T} \lambda^{|T|} (1-\lambda)^{|S-T|} \widehat{h}(S)}_{\widehat{M_\lambda h}(T)}, \tag{33}$$

where the final step follows by noting that each $\widehat{M_\lambda h}(T)$ depends only on $\widehat{h}(S)$ for $S \supseteq T$. $\qquad\square$

The expression derived in Lemma A.3 shows how spectral mass gets redistributed from higher-order to lower-order terms. To understand how smoothing affects classifier robustness, it is helpful to quantify how much of the original function's complexity (i.e., higher-order interactions) survives after smoothing. The following result shows how smoothing suppresses higher-order interactions by bounding how much mass survives in terms of degree $\geq k$.

**Theorem A.4** (Higher-order Spectral Mass After Smoothing). *For any function* $h : \{0,1\}^n \to \mathbb{R}$, *smoothing parameter* $\lambda \in [0,1]$, *and* $0 \leq k \leq n$,

$$\sum_{T : |T| \geq k} |\widehat{M_\lambda h}(T)| \leq \Pr_{X \sim \mathrm{Bin}(n,\lambda)}[X \geq k] \sum_{S : |S| \geq k} |\widehat{h}(S)|. \tag{34}$$

*Proof.* We first apply Lemma A.3 to expand each $\widehat{M_\lambda h}(T)$ and derive

$$\sum_{T : |T| \geq k} |\widehat{M_\lambda h}(T)| \leq \sum_{T : |T| \geq k} \sum_{S \supseteq T} \lambda^{|T|} (1-\lambda)^{|S-T|} |\widehat{h}(S)| \tag{35}$$

$$= \sum_{S : |S| \geq k} |\widehat{h}(S)| \underbrace{\sum_{j=k}^{|S|} \binom{|S|}{j} \lambda^j (1-\lambda)^{|S|-j}}_{\Pr_{Y \sim \mathrm{Bin}(|S|,\lambda)}[Y \geq k]} \tag{36}$$

where we re-indexed the summations to track the contribution of each $|\widehat{h}(S)|$ for $|S| \geq k$. To yield the desired result, we next apply the following inequality of binomial tail CDFs given $|S| \leq n$:

$$\Pr_{Y \sim \mathrm{Bin}(|S|,\lambda)}[Y \geq k] \leq \Pr_{X \sim \mathrm{Bin}(n,\lambda)}[X \geq k]. \tag{37}$$

$$\square$$

Our analyses with respect to the standard basis provide a first step towards understanding the random masking operator $M_\lambda$. However, the weight-mixing from our initial calculations suggests that the standard basis may be algebraically challenging to work with.

### A.3 Analysis in the Biased Fourier Basis

While analysis on the standard Fourier basis reveals interesting properties about $M_\lambda$, it suggests that this may not be the natural choice of basis in which to analyze random masking. Principally, this is because each $M_\lambda \chi_S$ is expressed as a linear combination of $\chi_T$ where $T \subseteq S$. By "natural", we instead aim to express the image of $M_\lambda$ as a single term. One partial attempt is an extension of the standard basis, known as the $p$-biased basis, which is defined as follows.

**Definition A.5** (*p*-Biased Basis). For each subset $S \subseteq [n]$, define its *p*-biased function basis as:

$$\chi_S^p(\alpha) = \prod_{i \in S} \frac{p - \alpha_i}{\sqrt{p - p^2}}. \tag{38}$$

Observe that when $p = 1/2$, this is the standard basis discussed earlier. The *p*-biased basis is orthonormal with respect to the *p*-biased distribution on $\{0,1\}^n$ in that:

$$\mathop{\mathbb{E}}_{\alpha \sim \mathrm{Bern}(p)^n} [\chi_S^p(\alpha)\chi_T^p(\alpha)] = \begin{cases} 1 & \text{if } S = T, \\ 0 & \text{if } S \neq T. \end{cases} \tag{39}$$

On the *p*-biased basis, smoothing with a well-chosen $\lambda$ induces a change-of-basis effect.

**Lemma A.6** (Change-of-Basis). *For any p-biased basis function $\chi_S^p$ and $\lambda \in [p, 1]$,*

$$M_\lambda \chi_S^p(\alpha) = \left(\frac{\lambda - p}{1 - p}\right)^{|S|/2} \chi_S^{p/\lambda}(\alpha). \tag{40}$$

*Proof.* Expanding the definition of $M_\lambda$, we first derive:

$$M_\lambda \chi_S^p(\alpha) = \mathop{\mathbb{E}}_{z \sim \mathrm{Bern}(\lambda)^n} \left[ \prod_{i \in S} \frac{p - \alpha_i z_i}{\sqrt{p - p^2}} \right] \tag{41}$$

$$= \prod_{i \in S} \mathop{\mathbb{E}}_z \left[ \frac{p - \alpha_i z_i}{\sqrt{p - p^2}} \right] \qquad \text{(by independence of } z_1, \ldots, z_n\text{)}$$

$$= \prod_{i \in S} \frac{p - \lambda \alpha_i}{\sqrt{p - p^2}}, \tag{42}$$

We then rewrite the above in terms of a $(p/\lambda)$-biased basis function as follows:

$$M_\lambda \chi_S^p(\alpha) = \prod_{i \in S} \lambda \frac{(p/\lambda) - \alpha_i}{\sqrt{p - p^2}} \tag{43}$$

$$= \prod_{i \in S} \lambda \frac{\sqrt{(p/\lambda) - (p/\lambda)^2}}{\sqrt{p - p^2}} \frac{(p/\lambda) - \alpha_i}{\sqrt{(p/\lambda) - (p/\lambda)^2}} \qquad (\lambda \geq p)$$

$$= \prod_{i \in S} \sqrt{\frac{\lambda - p}{1 - p}} \frac{(p/\lambda) - \alpha_i}{\sqrt{(p/\lambda) - (p/\lambda)^2}} \tag{44}$$

$$= \left(\frac{\lambda - p}{1 - p}\right)^{|S|/2} \underbrace{\prod_{i \in S} \frac{(p/\lambda) - \alpha_i}{\sqrt{(p/\lambda) - (p/\lambda)^2}}}_{\chi_S^{p/\lambda}(\alpha)} \tag{45}$$

$\square$

When measured with respect to this changed basis, $M_\lambda$ provably contracts the variance.

**Theorem A.7** (Variance Reduction). *For any function $h : \{0,1\}^n \to \mathbb{R}$ and $\lambda \in [p, 1]$,*

$$\mathop{\mathrm{Var}}_{\alpha \sim \mathrm{Bern}(p/\lambda)^n} [M_\lambda h(\alpha)] \leq \left(\frac{\lambda - p}{1 - p}\right) \mathop{\mathrm{Var}}_{\alpha \sim \mathrm{Bern}(p)^n} [h(\alpha)]. \tag{46}$$

*If the function is centered at $\mathbb{E}_{\alpha \sim \mathrm{Bern}(p)^n}[h(\alpha)] = 0$, then we also have:*

$$\mathop{\mathbb{E}}_{\alpha \sim \mathrm{Bern}(p/\lambda)^n} \left[M_\lambda h(\alpha)^2\right] \leq \mathop{\mathbb{E}}_{\alpha \sim \mathrm{Bern}(p)} \left[h(\alpha)^2\right]. \tag{47}$$

*Proof.* We use the previous results to compute:

$$\operatorname*{Var}_{\alpha\sim\text{Bern}(p/\lambda)^n}[M_\lambda h(\alpha)] = \operatorname*{Var}_{\alpha\sim\text{Bern}(p/\lambda)^n}\left[M_\lambda \sum_{S\subseteq[n]} \widehat{h}(S)\chi_S^p(\alpha)\right]$$

(by unique $p$-biased representation of $h$)

$$= \operatorname*{Var}_{\alpha\sim\text{Bern}(p/\lambda)^n}\left[\sum_{S\subseteq[n]}\left(\frac{\lambda-p}{1-p}\right)^{|S|/2}\widehat{h}(S)\chi_S^{p/\lambda}(\alpha)\right]$$

(by linearity and Lemma A.6)

$$= \sum_{S\neq\emptyset}\left(\frac{\lambda-p}{1-p}\right)^{|S|}\widehat{h}(S)^2 \qquad \text{(Parseval's by orthonormality of } \chi_S^{p/\lambda})$$

$$\leq \left(\frac{\lambda-p}{1-p}\right)\sum_{S\neq\emptyset}\widehat{h}(S)^2 \qquad (0\leq\tfrac{\lambda-p}{1-p}\leq 1 \text{ because } p\leq\lambda\leq 1)$$

$$= \left(\frac{\lambda-p}{1-p}\right)\operatorname*{Var}_{\alpha\sim\text{Bern}(p)^n}[h(\alpha)] \qquad \text{(Parseval's by orthonormality of } \chi_S^p)$$

leading to the first desired inequality. For the second inequality, we continue from the above to get:

$$\mathbb{E}_{\alpha\sim\text{Bern}(p)^n}[h(\alpha)^2] = \widehat{h}(\emptyset)^2 + \underbrace{\sum_{S\neq\emptyset}\widehat{h}(S)^2}_{\text{Var}[h(\alpha)]}, \tag{48}$$

$$\mathbb{E}_{\alpha\sim\text{Bern}(p/\lambda)^n}[M_\lambda h(\alpha)^2] = \widehat{M_\lambda h}(\emptyset)^2 + \underbrace{\sum_{S\neq\emptyset}\widehat{M_\lambda h}(S)^2}_{\text{Var}[M_\lambda h(\alpha)]}, \tag{49}$$

where recall that $\widehat{h}(\emptyset) = \mathbb{E}_\alpha[h(\alpha)] = 0$ by assumption. $\qquad\square$

The smoothing operator $M_\lambda$ acts like a downshift on the standard basis and as a change-of-basis on a well-chosen $p$-biased basis. In both cases, the algebraic manipulations can be cumbersome and inconvenient, suggesting that neither is the natural choice of basis for studying $M_\lambda$. To address this limitation, we use the monotone basis in Appendix B to provide a novel and tractable characterization of how smoothing affects the spectrum and stability of Boolean functions.

# B    Analysis of Stability and Smoothing in the Monotone Basis

While the standard Fourier basis is a common starting point for studying Boolean functions, its interaction with $M_\lambda$ is algebraically complex. The main reason is that the Fourier basis treats $0\to 1$ and $1\to 0$ perturbations symmetrically. In contrast, we wish to analyze perturbations that add features (i.e., $\alpha'\sim\Delta_r(\alpha)$) and smoothing operations that remove features. This mismatch results in a complex redistribution of terms that is algebraically inconvenient to manipulate. We were thus motivated to adopt the *monotone basis* (also known as unanimity functions in game theory), under which smoothing by $M_\lambda$ is well-behaved.

## B.1    Monotone Basis for Boolean Functions

For any subset $T\subseteq[n]$, define its corresponding *monotone basis function* $\mathbf{1}_T : \{0,1\}^n \to \{0,1\}$ as:

$$\mathbf{1}_T(\alpha) = \begin{cases} 1 & \text{if } \alpha_i = 1 \text{ for all } i\in T \text{ (all features in } S \text{ present)}, \\ 0 & \text{otherwise (any feature in } T \text{ is absent)}, \end{cases} \tag{50}$$

where let $\mathbf{1}_\emptyset(\alpha) = 1$. First, we flexibly identify subsets of $[n]$ with binary vectors in $\{0,1\}^n$, which lets us write $T\subseteq\alpha$ if $i\in T$ implies $\alpha_i = 1$. This gives us useful ways to equivalently write $\mathbf{1}_T(\alpha)$:

$$\mathbf{1}_T(\alpha) = \prod_{i\in T}\alpha_i = \begin{cases} 1 & \text{if } T\subseteq\alpha, \\ 0 & \text{otherwise.} \end{cases} \tag{51}$$

The monotone basis lets us more compactly express properties that depend on the inclusion or exclusion of features. For instance, the earlier example of conjunction $h(\alpha) = \alpha_1 \wedge \alpha_2$ may be equivalently written as:

$$\alpha_1 \wedge \alpha_2 = \mathbf{1}_{\{1,2\}}(\alpha) \qquad \text{(monotone basis)}$$

$$= \frac{1}{4}\chi_\emptyset(\alpha) - \frac{1}{4}\chi_{\{1\}}(\alpha) - \frac{1}{4}\chi_{\{2\}}(\alpha) + \frac{1}{4}\chi_{\{1,2\}}(\alpha) \qquad \text{(standard basis)}$$

Unlike the standard bases (both standard Fourier and $p$-biased Fourier), the monotone basis is not orthonormal with respect to $\{0,1\}^n$ because

$$\mathop{\mathbb{E}}_{\alpha \sim \{0,1\}^n}\left[\mathbf{1}_S(\alpha)\mathbf{1}_T(\alpha)\right] = \mathop{\Pr}_{\alpha \sim \{0,1\}^n}\left[S \cup T \subseteq \alpha\right] = 2^{-|S \cup T|}, \tag{52}$$

where note that $S \cup T \subseteq \alpha$ iff both $S \subseteq \alpha$ and $T \subseteq \alpha$. However, the monotone basis does satisfy some interesting properties, which we describe next.

**Theorem B.1.** *Any function $h : \{0,1\}^n \to \mathbb{R}^n$ is uniquely expressible in the monotone basis as:*

$$h(\alpha) = \sum_{T \subseteq [n]} \widetilde{h}(T)\mathbf{1}_T(\alpha), \tag{53}$$

*where $\widetilde{h}(T) \in \mathbb{R}$ are the monotone basis coefficients of $h$ that can be recursively computed via:*

$$\widetilde{h}(T) = h(T) - \sum_{S \subsetneq T} \widetilde{h}(S), \quad \widetilde{h}(\emptyset) = h(\mathbf{0}_n), \tag{54}$$

*where $h(T)$ denotes the evaluation of $h$ on the binary vectorized representation of $T$.*

*Proof.* We first prove existence and uniqueness. By definition of $\mathbf{1}_T$, we have the simplification:

$$h(\alpha) = \sum_{T \subseteq [n]} \widetilde{h}(T)\mathbf{1}_T(\alpha) = \sum_{T \subseteq \alpha} \widetilde{h}(T). \tag{55}$$

This yields a system of $2^n$ linear equations (one for each $h(\alpha)$) in $2^n$ unknowns (one for each $\widetilde{h}(T)$). We may treat this as a matrix of size $2^n \times 2^n$ with rows indexed by $h(\alpha)$ and columns indexed by $\widetilde{h}(T)$, sorted by inclusion and degree. This matrix is lower-triangular with ones on the diagonal ($\mathbf{1}_T(T) = 1$ and $\mathbf{1}_T(\alpha) = 0$ for $|T| > \alpha$; like a transposed Equation (29)), and so the $2^n$ values of $h(\alpha)$ uniquely determine $\widetilde{h}(T)$.

For the recursive formula, we simultaneously substitute $\alpha \mapsto T$ and $T \mapsto S$ in Equation (55) to write:

$$h(T) = \widetilde{h}(T) + \sum_{S \subsetneq T} \widetilde{h}(S), \tag{56}$$

and re-ordering terms yields the desired result. $\qquad\qquad\qquad\qquad\qquad\qquad\qquad\qquad\qquad \square$

## B.2 Smoothing and Stability in the Monotone Basis

A key advantage of the monotone basis is that it yields a convenient analytical expression for how smoothing affects the spectrum.

**Theorem B.2** (Smoothing in the Monotone Basis). *Let $M_\lambda$ be the smoothing operator as in Definition A.1. Then, for any function $h : \{0,1\}^n \to \mathbb{R}$ and $T \subseteq [n]$, we have the spectral contraction:*

$$\widetilde{M_\lambda h}(T) = \lambda^{|T|}\widetilde{h}(T),$$

*where $\widetilde{M_\lambda h}(T)$ and $\widetilde{h}(T)$ are the monotone basis coefficients of $M_\lambda h$ and $h$ at $T$, respectively.*

*Proof.* By linearity of expectation, it suffices to study how $M_\lambda$ acts on each basis function:

$$M_\lambda \mathbf{1}_T(\alpha) = \underset{z \sim \text{Bern}(\lambda)^n}{\mathbb{E}} \left[ \mathbf{1}_T(\alpha \odot z) \right] \qquad \text{(by definition of } M_\lambda)$$

$$= \underset{z \sim \text{Bern}(\lambda)^n}{\mathbb{E}} \left[ \prod_{i \in T} (\alpha_i z_i) \right] \qquad \text{(by definition of } \mathbf{1}_T(\alpha))$$

$$= \prod_{i \in T} \left( \alpha_i \underset{z_i \sim \text{Bern}(\lambda)}{\mathbb{E}} [z_i] \right) \qquad \text{(by independence of } z_1, \ldots, z_n)$$

$$= \lambda^{|T|} \mathbf{1}_T(\alpha) \qquad (\mathbb{E}[z_i] = \lambda)$$

$\square$

The monotone basis also gives a computationally tractable way of bounding the stability rate. Crucially, the difference between two Boolean functions is easier to characterize. As a simplified setup, we consider classifiers of form $h : \{0,1\}^n \to \mathbb{R}$, where for $\beta \sim \Delta_r(\alpha)$ let:

$$h(\beta) \cong h(\alpha) \quad \text{if} \quad |h(\beta) - h(\alpha)| \le \gamma. \tag{57}$$

Such $h$ and its decision boundary $\gamma$ may be derived from a general classifier $f : \mathbb{R}^n \to \mathbb{R}^m$ once $x$ and $\alpha$ are known. This relation of the decision boundary then motivates the difference computation:

$$h(\beta) - h(\alpha) = \sum_{T \subseteq [n]} \widetilde{h}(T)(\mathbf{1}_T(\beta) - \mathbf{1}_T(\alpha)) = \sum_{T \subseteq \beta \setminus \alpha, T \neq \emptyset} \widetilde{h}(T), \tag{58}$$

where recall that $\mathbf{1}_T(\beta) - \mathbf{1}_T(\alpha) = 1$ iff $T \neq \emptyset$ and $T \subseteq \beta \setminus \alpha$. This algebraic property plays a key role in tractably bounding the stability rate. Specifically, we upper-bound the *instability rate* $1 - \tau_r$:

$$1 - \tau_r = \underset{\beta \sim \Delta_r(\alpha)}{\text{Pr}} [|h(\beta) - h(\alpha)| > \gamma]. \tag{59}$$

An upper bound of form $1 - \tau_r \le Q$, where $Q$ depends on the monotone coefficients of $h$, then implies a lower bound on the stability rate $1 - Q \le \tau_r$. We show this next.

**Lemma B.3** (Stability Rate Bound). *For any function $h : \{0,1\}^n \to [0,1]$ and attribution $\alpha \in \{0,1\}^n$ that satisfy Equation (57), the stability rate $\tau_r$ is bounded by:*

$$1 - \tau_r \le \frac{1}{\gamma} \sum_{k=1}^r \sum_{\substack{T \subseteq [n] \setminus \alpha \\ |T| = k}} |\widetilde{h}(T)| \cdot \underset{\beta \sim \Delta_r}{\text{Pr}} [|\beta \setminus \alpha| \ge k], \tag{60}$$

*where*

$$\underset{\beta \sim \Delta_r}{\text{Pr}} [|\beta \setminus \alpha| \ge k] = \frac{1}{|\Delta_r|} \sum_{j=k}^r \binom{n - |\alpha| - k}{j - k}, \quad |\Delta_r| = \sum_{i=0}^r \binom{n - |\alpha|}{i} \tag{61}$$

*Proof.* We can directly bound the stability rate as follows:

$$1 - \tau_r = \Pr_{\beta \sim \Delta_r} [|h(\beta) - h(\alpha)| > \gamma] \tag{62}$$

$$\leq \frac{1}{\gamma} \underset{\beta \sim \Delta_r}{\mathbb{E}} [|h(\beta) - h(\alpha)|] \qquad \text{(Markov's inequality)}$$

$$\leq \frac{1}{\gamma} \underset{\beta \sim \Delta_r}{\mathbb{E}} \sum_{\substack{T \subseteq \beta \setminus \alpha \\ T \neq \emptyset}} |\widetilde{h}(T)| \qquad \text{(by Equation (58), triangle inequality)}$$

$$= \frac{1}{\gamma |\Delta_r|} \sum_{k=0}^{r} \sum_{|\beta \setminus \alpha| = k} \sum_{\substack{T \subseteq \beta \setminus \alpha \\ T \neq \emptyset}} |\widetilde{h}(T)| \qquad \text{(enumerate } \beta \in \Delta_r(\alpha) \text{ by its size, } k)$$

$$= \frac{1}{\gamma |\Delta_r|} \sum_{k=1}^{r} \sum_{\substack{S \subseteq [n] \setminus \alpha \\ |S| = k}} \sum_{\substack{T \subseteq S \\ T \neq \emptyset}} |\widetilde{h}(T)| \qquad \text{(the } k = 0 \text{ term is zero, and let } S = \beta \setminus \alpha)$$

$$= \frac{1}{\gamma |\Delta_r|} \sum_{k=1}^{r} \sum_{\substack{T \subseteq [n] \setminus \alpha \\ |T| = k}} |\widetilde{h}(T)| \cdot \underbrace{|\{S \subseteq [n] \setminus \alpha : S \supseteq T, |S| \leq r\}|}_{\text{Total times that } \widetilde{h}(T) \text{ appears}} \qquad \text{(re-index by } T)$$

$$= \frac{1}{\gamma} \sum_{k=1}^{r} \sum_{\substack{T \subseteq [n] \setminus \alpha \\ |T| = k}} |\widetilde{h}(T)| \cdot \Pr_{\beta \sim \Delta_r} [|\beta \setminus \alpha| \geq k] \tag{63}$$

$\square$

An immediate consequence from Theorem B.2 is a stability rate bound on smoothed functions.

**Theorem B.4** (Stability of Smoothed Functions). *Consider any function $h : \{0,1\}^n \to [0,1]$ and attribution $\alpha \in \{0,1\}^n$ that satisfy Equation (57). Then, for any $\lambda \in [0,1]$,*

$$1 - \frac{Q}{\gamma} \leq \tau_r(h, \alpha) \implies 1 - \frac{\lambda Q}{\gamma} \leq \tau_r(M_\lambda h, \alpha), \tag{64}$$

*where*

$$Q = \sum_{k=1}^{r} \sum_{\substack{T \subseteq [n] \setminus \alpha \\ |T| = k}} |\widetilde{h}(T)| \cdot \Pr_{\beta \sim \Delta_r} [|\beta \setminus \alpha| \geq k]. \tag{65}$$

*Proof.* This follows from applying Theorem B.2 to Lemma B.3 by noting that:

$$1 - \tau_r(M_\lambda h, \alpha) \leq \frac{1}{\gamma} \sum_{k=1}^{r} \lambda^k \sum_{\substack{T \subseteq [n] \setminus \alpha \\ |T| = k}} |\widetilde{h}(T)| \cdot \Pr_{\beta \sim \Delta_r} [|\beta \setminus \alpha| \geq k]. \tag{66}$$

$\square$

Moreover, we also present the following result on hard stability in the monotone basis.

**Theorem B.5** (Hard Stability Bound). *For any function $h : \{0,1\}^n \to [0,1]$ and attribution $\alpha \in \{0,1\}^n$ that satisfy Equation (57), let*

$$r^\star = \arg\max_{r \geq 0} \max_{\beta : |\beta \setminus \alpha| \leq r} \left[ \left| \sum_{T \subseteq \beta \setminus \alpha, T \neq \emptyset} \widetilde{h}(T) \right| \leq \gamma \right]. \tag{67}$$

*Then, $h$ is hard stable at $\alpha$ with radius $r^\star$.*

*Proof.* This follows from Equation (58) because it is equivalent to stating that:

$$r^\star = \arg\max_{r \geq 0} \max_{\beta : |\beta \setminus \alpha| \leq r} \underbrace{[|h(\beta) - h(\alpha)| \leq \gamma]}_{h(\beta) \cong h(\alpha)}. \tag{68}$$

$\square$

In summary, the monotone basis provides a more natural setting in which to study the smoothing operator $M_\lambda$. While $M_\lambda$ yields an algebraically complex weight redistribution under the standard basis, its effect is more compactly described in the monotone basis as a point-wise contraction at each $T \subseteq [n]$. In particular, we are able to derive a lower-bound improvement on the stability of smoothed functions in Theorem B.4.

## C   Additional Experiments

In this section, we include experiment details and additional experiments.

**Models**   For vision models, we used Vision Transformer (ViT) [17], ResNet50, and ResNet18 [24]. For language models, we used RoBERTa [38].

**Datasets**   For the vision dataset, we used a subset of ImageNet that contains two images per class, for a total of 2000 images. The images are of size $3 \times 224 \times 224$, which we segmented into grids with patches of size $16 \times 16$, for a total of $n = (224/16)^2 = 196$ features. For the language dataset, we used six subsets of TweetEval (emoji, emotion, hate, irony, offensive, sentiment) for a total of 10653 items; we omitted the stance subset because their corresponding fine-tuned models were not readily available.

**Explanation Methods**   For feature attribution methods, we used LIME [51], SHAP [39], Integrated Gradients [63], and MFABA [75] using the implementation from exlib. [7] Each attribution method outputs a ranking of features by their importance score, which we binarized by selecting the top-25% of features.

**Certifying Stability with SCA**   We used SCA (Equation (3)) for certifying soft stability (Theorem 3.1) with parameters of $\varepsilon = \delta = 0.1$, for a sample size of $N = 150$. We use the same $N$ when certifying hard stability via SCA-hard (Theorem 3.2). Stability rates for shorter text sequences were right-padded by repeating their final value. Where appropriate, we used 1000 iterations of bootstrap to compute the 95% confidence intervals.

**Compute**   We used a cluster with NVIDIA GeForce RTX 3090 and NVIDIA RTX A6000 GPUs.

### C.1   Certifying Hard Stability with MuS

We next discuss how Xue et al. [70] compute hard stability certificates with MuS-smoothed classifiers.

**Theorem C.1** (Certifying Hard Stability via MuS [70]). *For any classifier $f : \mathbb{R}^n \to [0,1]^m$ and $\lambda \in [0,1]$, let $\tilde{f} = M_\lambda f$ be the MuS-smoothed classifier. Then, for any input $x \in \mathbb{R}^n$ and explanation $\alpha \in \{0,1\}^n$, the certifiable hard stability radius is given by:*

$$r_{\text{cert}} = \frac{1}{2\lambda} \left[ \tilde{f}_1(x \odot \alpha) - \tilde{f}_2(x \odot \alpha) \right], \tag{69}$$

*where $\tilde{f}_1(x \odot \alpha)$ and $\tilde{f}_2(x \odot \alpha)$ are the top-1 and top-2 class probabilities of $\tilde{f}(x \odot \alpha)$.*

Each output coordinate $\tilde{f}_1, \ldots, \tilde{f}_m$ is also $\lambda$-Lipschitz to the masking of features:

$$|\tilde{f}_i(x \odot \alpha) - \tilde{f}_i(x \odot \alpha')| \leq \lambda |\alpha - \alpha'|, \quad \text{for all } \alpha, \alpha' \in \{0,1\}^n \text{ and } i = 1, \ldots, m. \tag{70}$$

That is, the keep-probability of each feature is also the Lipschitz constant (per earlier discussion: $\kappa = \lambda$). Note that deterministically evaluating $M_\lambda f_x$ would require $2^n$ samples in total, as there

---

[7] https://github.com/BrachioLab/exlib

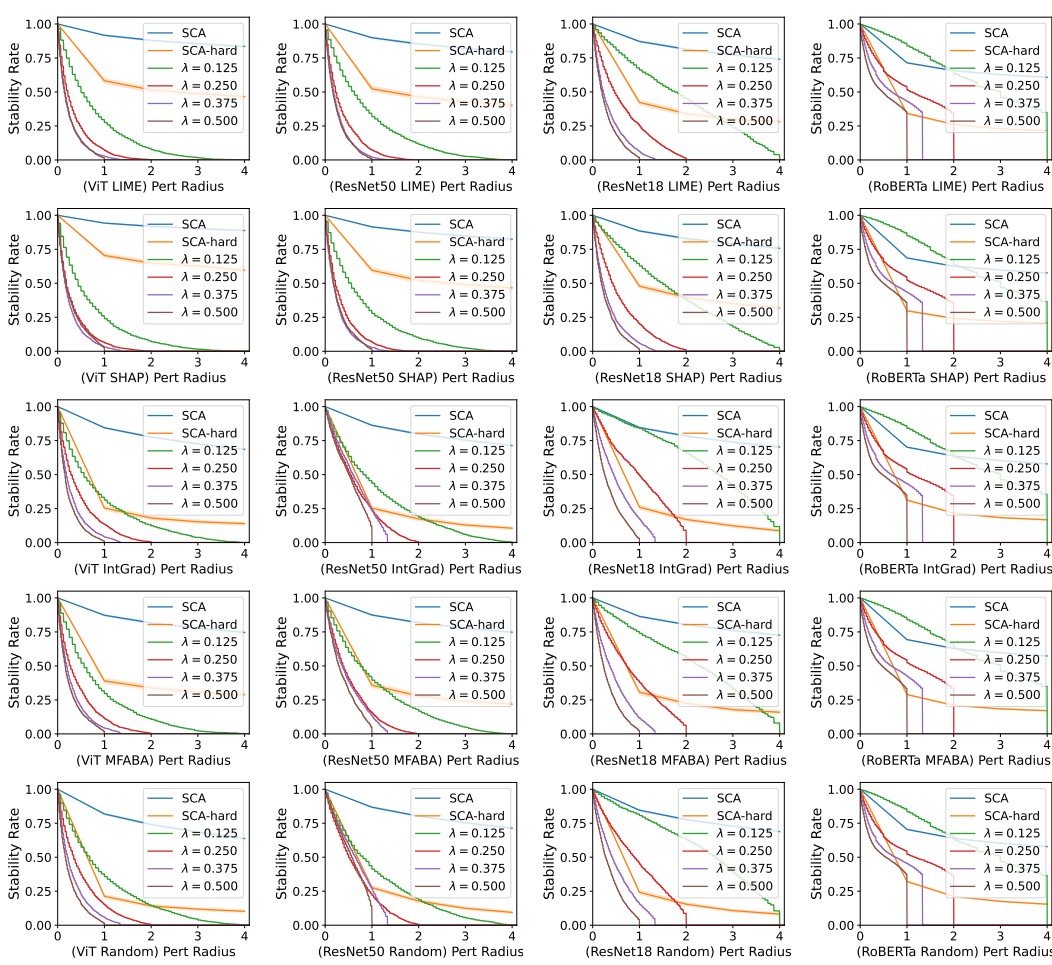

Figure 9: **SCA certifies more than MuS.** An extended version of Figure 5. SCA-based stability guarantees are typically much stronger than those from MuS.

are $2^n$ possibilities for $\text{Bern}(\lambda)^n$. Interestingly, distributions other than $\text{Bern}(\lambda)^n$ also suffice to attain the desired Lipschitz constant, and thus a hard stability certificate. In fact, Xue et al. [70] constructs such a distribution based on de-randomized sampling [34], for which a smoothed classifier is deterministically computed in $\ll 2^n$ samples. However, our Boolean analytic results do not readily extend to non-Bernoulli distributions.

## C.2  SCA vs. MuS on Different Explanation Methods

We show in Figure 9 an extension of Figure 5, where we include all explanation methods. Similar to the main paper, we observe that SCA typically obtains stronger stability certificates than MuS, especially on vision models. On RoBERTa, MuS certificates can be competitive for small radii, but this requires a very smooth classifier ($\lambda = 0.125$).

## C.3  MuS-based Hard Stability Certificates

We show in Figure 10 that MuS-based certificates struggle to distinguish between explanation methods. This is in contrast to SCA-based certificates, which show that LIME and SHAP tend to be more stable. The plots shown here contain the same information as previously presented in Figure 9, except that we group the data by model and certification method.

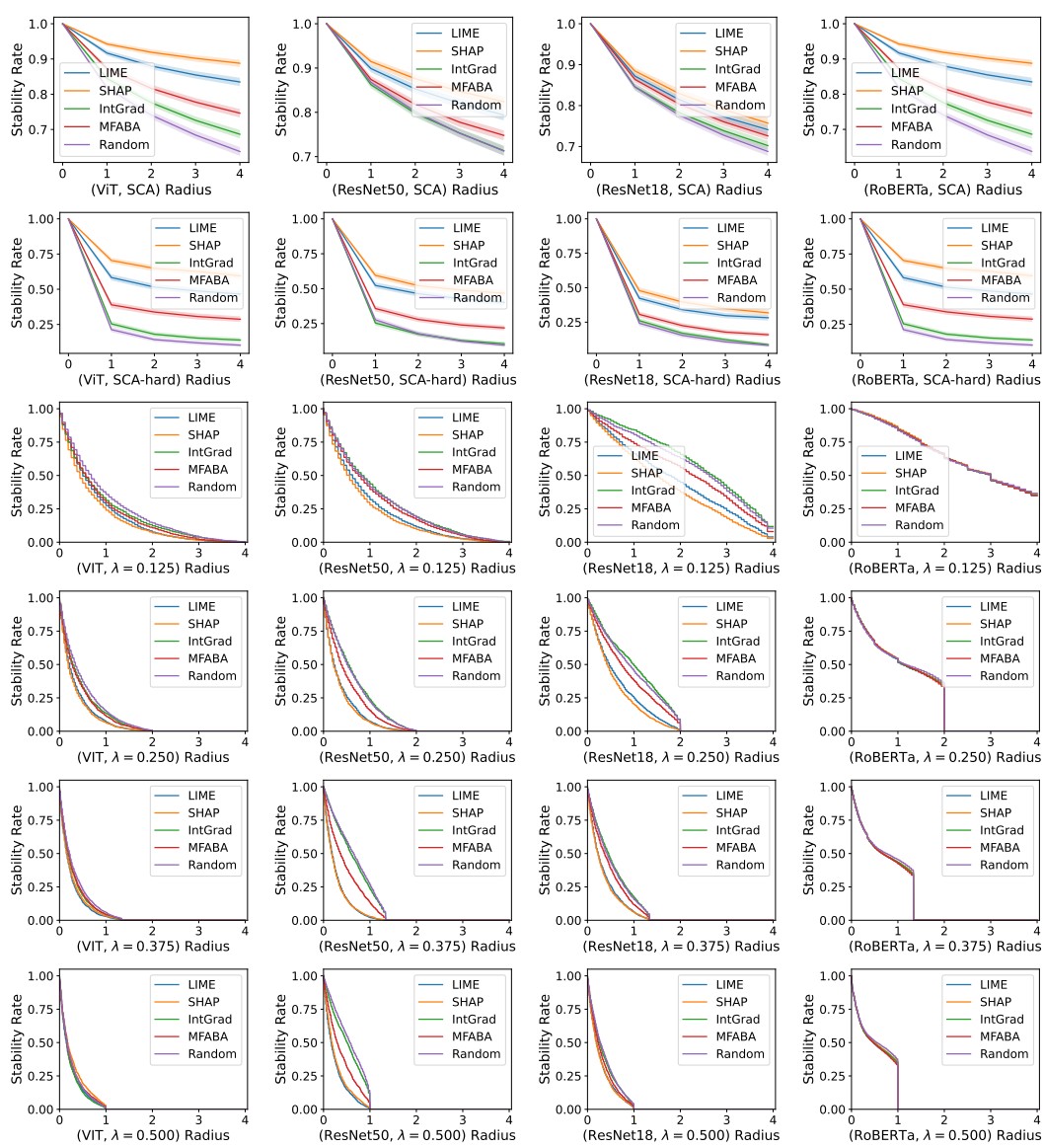

Figure 10: **MuS-based hard stability struggles to distinguish explanation methods.** SCA-based stability certificates (top two rows) show that LIME and SHAP tend to be the most stable.

### C.4  Ablation on Top-k Feature Selection

To see the stability of explanation methods across different selections of top-$k$, we show an ablation study in Figure 11. Notably, we observe that SHAP is generally the most stable, whereas Integrated Gradients and the random baseline tend to be the least stable.

### C.5  Stability vs. Smoothing

We show in Figure 12 an extension of Figure 7, where we plot perturbations at larger radii. While stability trends extend to larger radii, the effect is most pronounced at smaller radii. Nevertheless, even mild smoothing yields benefits at radii beyond what MuS can reasonably certify without significantly degrading accuracy.

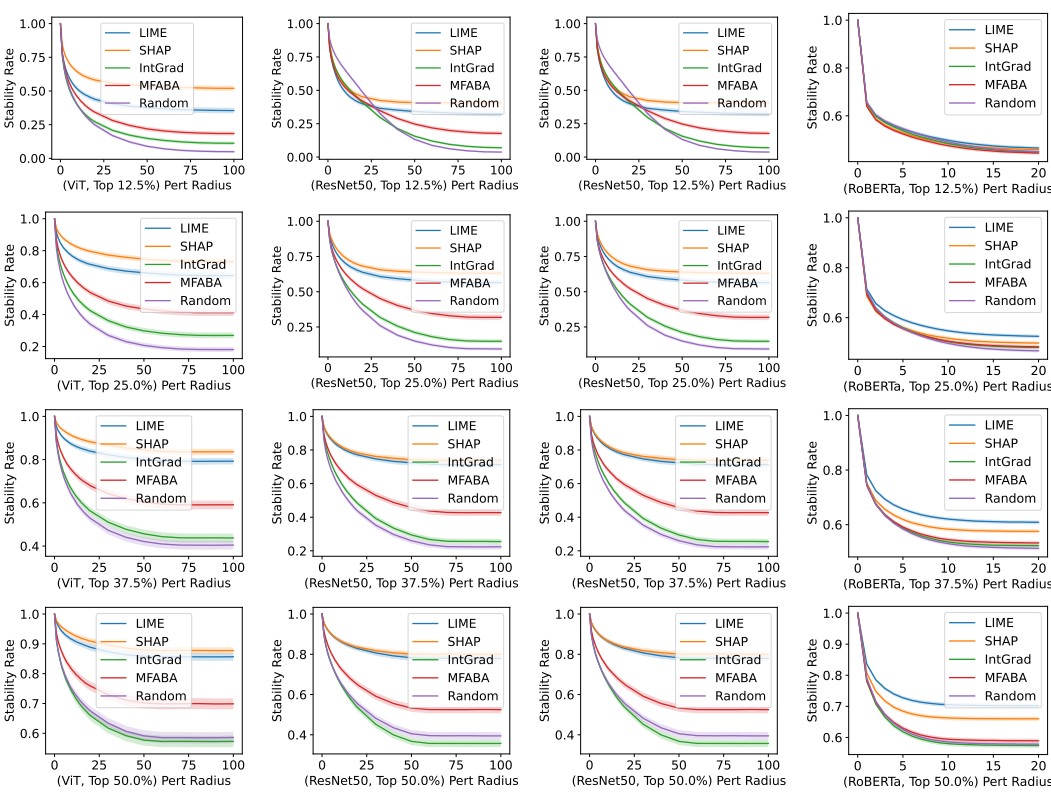

Figure 11: **Soft stability rates on different top-$k$ selection.** SHAP tends to be the most stable method, particularly for vision models. On the other hand, Integrated Gradients and the random baseline are usually the least stable. Note that the top-$25\%$ row of plots is the same one as shown earlier in Figure 6.

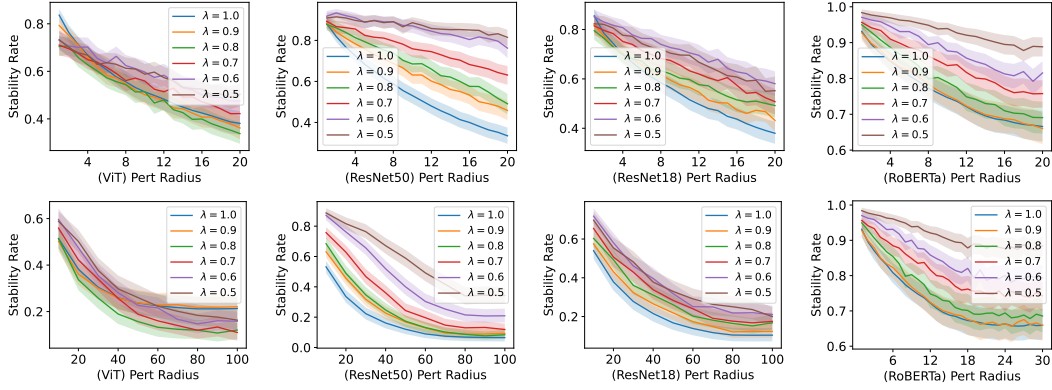

Figure 12: **Mild smoothing ($\lambda \geq 0.5$) can improve stability.** An extended version of Figure 7. The improvement is more pronounced at smaller radii (top row) than at larger radii (bottom row).

### C.6 Computational Efficiency of Certification

Certifying soft stability requires $\frac{\log(2/\delta)}{2\varepsilon^2}$ forward passes of the model. However, the exact wall-clock time depends on system and implementation-specific details. In particular, batched evaluation of the samples can speed up the individual per-sample forward pass, as we show in Table 1 using different batch sizes. We report statistics for each model averaged over 100 samples from its respective dataset.

| | Baseline | Effective Time per Pass (ms) with Batching | | |
|---|---|---|---|---|
| Model | Time (ms) | Batch Size 5 | Batch Size 10 | Batch Size 15 |
| ViT | $3.94 \pm 0.17$ | $1.60 \pm 0.06 \ (2.46\times)$ | $1.44 \pm 0.05 \ (2.74\times)$ | $1.46 \pm 0.06 \ (2.71\times)$ |
| ResNet50 | $3.82 \pm 0.10$ | $0.84 \pm 0.07 \ (4.52\times)$ | $0.48 \pm 0.08 \ (7.99\times)$ | $0.40 \pm 0.01 \ (9.57\times)$ |
| ResNet18 | $1.62 \pm 0.12$ | $0.39 \pm 0.04 \ (4.10\times)$ | $0.24 \pm 0.01 \ (6.72\times)$ | $0.19 \pm 0.01 \ (8.50\times)$ |
| RoBERTa | $4.81 \pm 0.14$ | $3.84 \pm 0.10 \ (1.25\times)$ | $3.66 \pm 0.09 \ (1.31\times)$ | $3.77 \pm 0.13 \ (1.28\times)$ |

Table 1: **Batching significantly reduces the effective time per forward pass.** We compare the baseline single-pass time against the effective per-pass time achieved during stability certification (which requires $N = 150$ passes for $\varepsilon = \delta = 0.1$). The speedup factor relative to the baseline is shown in parentheses.

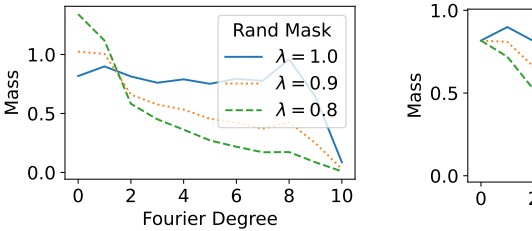

Figure 13: **Random masking and flipping are fundamentally different.** On the standard Fourier spectrum, random masking (left) causes a down-shift in spectral mass, where note that the orange and green curves are higher than the blue curve at lower degrees. In contrast, the more commonly studied random flipping (right) causes a point-wise contraction: the curve with smaller $\lambda$ is always lower.

### C.7 Random Masking vs. Random Flipping

We next study how the Fourier spectrum is affected by random masking and random flipping (i.e., the noise operator), which are respectively defined for Boolean functions as follows:

$$M_\lambda h(\alpha) = \mathop{\mathbb{E}}_{z \sim \text{Bern}(\lambda)^n} [h(\alpha \odot z)] \qquad \text{(random masking)}$$

$$T_\lambda h(\alpha) = \mathop{\mathbb{E}}_{z \sim \text{Bern}(q)^n} [h((\alpha + z) \bmod 2)], \quad q = \frac{1 - \lambda}{2} \qquad \text{(random flipping)}$$

In both cases, $\lambda \approx 1$ corresponds to mild smoothing, whereas $\lambda \approx 0$ corresponds to heavy smoothing. To study the difference between random masking and random flipping, we randomly generated a spectrum via $\widehat{h}(S) \sim N(0, 1)$ for each $S \subseteq [n]$. We then average the mass of the randomly masked and randomly flipped spectrum at each degree, which are respectively:

$$\text{Average mass at degree } k \text{ from random masking} = \sum_{S:|S|=k} |\widehat{M_\lambda h}(S)| \qquad (71)$$

$$\text{Average mass at degree } k \text{ from random flipping} = \sum_{S:|S|=k} |\widehat{T_\lambda h}(S)| \qquad (72)$$

We plot the results in Figure 13, which qualitatively demonstrates the effects of random masking and random flipping on the standard Fourier basis.

## D Additional Discussion

**Alternative Formulations of Stability** There are other ways to reasonably define stability. For example, one might define $\tau_{=k}$ as the probability that the prediction remains unchanged under an exactly $k$-sized additive perturbation. A conservative variant could then take the minimum over $\tau_{=1}, \ldots, \tau_{=r}$. The choice of formulation affects the implementation of the certification algorithm.

**SCA vs. MuS** While MuS offers deterministic (hard) guarantees, it is conservative and limited to small certified radii, making it less practical for distinguishing between feature attribution methods.

In contrast, SCA uses statistical methods to yield high-confidence probabilistic (soft) guarantees on the stability rate. More broadly, probabilistic guarantees are relevant for modern, large-scale systems as they are often more flexible and efficient than their deterministic counterparts. They have seen use in medical imaging [19], drug discovery [7], autonomous driving [37], and anomaly detection [35], often through conformal prediction [5, 8, 13].

**Limitations**    While soft stability provides a more fine-grained and model-agnostic robustness measure than hard stability, it remains sensitive to the choice of attribution thresholding and masking strategy. While standard, we only focus on square patches and top-25% selection. Additionally, our certificates are statistical rather than robustly adversarial, which may be insufficient in some high-stakes settings.

**Broader Impact**    Our work is useful for developing robust explanations for machine learning models. This would benefit practitioners who wish to gain a deeper understanding of model predictions. While our work may have negative impacts, it is not immediately apparent to us what they might be.

