# OpenReview forum: "Probabilistic Stability Guarantees for Feature Attributions"
_NeurIPS.cc/2025/Conference — NeurIPS 2025 poster_

### Official Review · Reviewer_SAAu · 2025-06-28

**Clarity:** 4
**Significance:** 3
**Originality:** 2
**Rating:** 5
**Confidence:** 3

**Summary:**

The authors present a notion of soft-stability for explanations: Given an explanation and a classifier prediction from masking features not included in the explanation, this captures the *probability* that the classifiers predictions are unchanged when masking using the set of explanations which include up to radius-$r$ more features than the original (where $r$ is a hyperparameter). This is an improvement over previously used hard-stability, which strictly labels an explanation as stable or unstable at a given radius. The authors demonstrate both qualitatively and quantitatively how their relaxed version of stability yields improvements over hard-stability. They propose an estimator for both hard and soft stability, for which they prove formal guarantees. Finally, the authors link stability and smoothing: smoother classifiers enjoying better worst-case stability.

**Questions:**

The first two questions are related to the weaknesses above

* Am I correct in my interpretation above that you are selecting some $k$ to go from real valued to binary explanations? If not, what are you doing to get the binary explanations? If you are: How do you select $k$? Is it a function of $r$, or chosen independently?
* While I mention many similarities with [1] I believe that your work still has novelty. Can you explain the similarities and differences with [1]? What does your method bring to the table that [1] does not? One difference is that the astuteness measure in [1] uses the $r$-ball around a point as opposed to your discrete masking. However major similarities, such as both providing a soft stability measure, remain. Would the method in [1] and this method provide similar results at similar radii? I.e., would their stability rankings for different methods agree? To this end, a variant of Figure 6 including the explainer astuteness measure from [1] could be valuable.
* In line 177 you define $\gamma$ as the “distance to the decision boundary”. Is this the distance of $x$ to the boundary? Or is it related to the distance of $x$ masked with $\alpha$ or $\alpha’$ to the boundary?

Based on the answers to these questions, I will consider raising my score by 1 point.

[1] Khan, Zulqarnain Q., et al. "Analyzing explainer robustness via probabilistic lipschitzness of prediction functions." International Conference on Artificial Intelligence and Statistics. PMLR, 2024.

**Ethical Concerns:**

["NO or VERY MINOR ethics concerns only"]

**Final Justification:**

I recommend acceptance for this paper. I maintain that this paper is clearly written and has a good mix of strong experiments and theory. The idea of relaxing hard stability to soft stability is intuitively important and the experimental results show the approach has clear value. I have raised my score from borderline accept to accept because my questions and weaknesses were well addressed. Namely the authors plan to make their binarization approach more explicit, show the method is useful at different $k$ via an ablation study, and effectively explain the differences with Khan et al. (2024).

**Limitations:**

yes

**Quality:**

3

**Strengths And Weaknesses:**

**Strengths**

* The paper is well written and visually appealing, with clear (and well-formatted) theorems and figures.
* The problem of providing a relaxed version of hard stability is important for obtaining a more nuanced understanding of trusting explanations.
* The estimator yields guarantees now for the classifier $f$ instead of smoothed $\hat{f}$, as in MuS.
* The experimental results are convincing
  * The SCA bound is indeed non-trivial at larger radii, whereas the MuS bound is.
  * Smoothing does seem to generally increase stability

**Weaknesses**

* You introduce explanations of feature $i$ as $\alpha_i \in \mathbb{R}$ and hence total explanations as $\alpha \in \mathbb{R}^N$ in section 2.1. However, you later move analysis to $\alpha \in \{0, 1\}^n$. You explain in line 66-67 that explanations can be converted into binary vectors via selecting the top-k. Two comments on this
  * (Minor) You never explicitly state that this is indeed what you are doing for the rest of the paper. Although that may be obvious to some, it would be beneficial to do so.
  * Such a step requires selecting some $k$ to use, but it is not made clear what $k$ is chosen or how. I can see such a choice being consequential.  For example, I imagine larger $k$ correlates with increased stability at all radii.
* There are two missing citations that would be beneficial for the paper
  * [1] This is a highly cited work on smoothing in interpretability, proposing the use of gaussian noise to smooth explanations. You mention such a practice in contrast to your discrete smoothing, and adding the citation there or in your related works section seems natural to me.
  * [2] This work has many similarities to yours. They propose *explainer astuteness* which takes in a radius $r$ and threshold $\lambda$ and returns the probability that explanations of points $x’$ in the $r$-ball around the point of interest $x$ are within $\lambda d(x,x’)$ of explanation for $x$. As a probability, explainer astuteness takes values from $0$ to $1$ and is thus a *soft measure of stability*. The authors also provide a link between model smoothness and their measure of stability: lower bounding explainer astuteness with the probabilistic lipschitzness [3] of the model: a relaxed version of local lipschitz.


[1] Smilkov, Daniel, et al. "Smoothgrad: removing noise by adding noise." arXiv preprint arXiv:1706.03825 (2017).

[2] Khan, Zulqarnain Q., et al. "Analyzing explainer robustness via probabilistic lipschitzness of prediction functions." International Conference on Artificial Intelligence and Statistics. PMLR, 2024.

[3] Mangal, Ravi, et al. "Probabilistic lipschitz analysis of neural networks." International Static Analysis Symposium. Cham: Springer International Publishing, 2020.

---

> ### Author Rebuttal · Authors · 2025-07-31
>
> We thank the reviewer for their thoughtful comments and helpful suggestions. Below, we address the main points raised.
>
> ## W1. Binarizing Explanations
> While we did describe the conversion from real to binary attributions at the end of Section 2.1, we acknowledge that this could be done more clearly and explicitly. We will accordingly improve the writing to improve exposition.
>
> To further clarify (re: Q1), we create a binary explanation mask $\alpha \in \\{0, 1\\}^n$ by selecting the top-k features of a real-valued vector in $\mathbb{R}^n$, a standard technique when evaluating feature attributions (Hooker et al., 2019). The hyperparameter k is chosen independently of the certification radius r. Our choice of k=25% in our experiments follows prior work in Xue et al. 2023.
>
>
>
> ## W2. Investigating Choice of k
>
> We follow the convention from previous work in Xue et al. (2023), which selects top k = 25% features as the explanation. However, the reviewer brings up a fair point that the choice of k may be consequential to our study. Therefore, to investigate further the influence of k, we performed a preliminary ablation study on varying k. The results do not fundamentally alter our original conclusions, though they show that stability tends to increase with k, which is expected. We will add this study to the appendix in the revised manuscript for further details.
>
> **ResNet18 with top-0.125 selection**
>
> | Radius | SHAP | LIME | IntGrad | MFABA | Rand |
> | ------ | ---- | ---- | ------- | ----- | ---- |
> | 0 | 1.000 | 1.000 | 1.000 | 1.000 | 1.000 |
> | 1 | 0.817 | 0.796 | 0.819 | 0.807 | 0.834 |
> | 2 | 0.734 | 0.705 | 0.742 | 0.726 | 0.756 |
> | 3 | 0.674 | 0.642 | 0.685 | 0.668 | 0.702 |
> | 4 | 0.625 | 0.595 | 0.640 | 0.623 | 0.656 |
> | 5 | 0.584 | 0.554 | 0.601 | 0.587 | 0.617 |
> | 6 | 0.552 | 0.521 | 0.567 | 0.554 | 0.583 |
> | 7 | 0.524 | 0.494 | 0.536 | 0.526 | 0.550 |
> | 8 | 0.501 | 0.469 | 0.507 | 0.503 | 0.523 |
> | 9 | 0.480 | 0.450 | 0.483 | 0.480 | 0.495 |
> | 10 | 0.461 | 0.433 | 0.458 | 0.458 | 0.473 |
>
>
> **Resnet18 with top-0.250 selection**
>
> | Radius | SHAP | LIME | IntGrad | MFABA | Rand |
> | ------ | ---- | ---- | ------- | ----- | ---- |
> | 0 | 1.000 | 1.000 | 1.000 | 1.000 | 1.000 |
> | 1 | 0.885 | 0.871 | 0.848 | 0.866 | 0.845 |
> | 2 | 0.830 | 0.813 | 0.783 | 0.806 | 0.778 |
> | 3 | 0.790 | 0.774 | 0.740 | 0.761 | 0.728 |
> | 4 | 0.757 | 0.740 | 0.701 | 0.727 | 0.691 |
> | 5 | 0.731 | 0.713 | 0.673 | 0.697 | 0.655 |
> | 6 | 0.707 | 0.691 | 0.647 | 0.672 | 0.627 |
> | 7 | 0.687 | 0.670 | 0.623 | 0.649 | 0.600 |
> | 8 | 0.670 | 0.653 | 0.601 | 0.629 | 0.578 |
> | 9 | 0.655 | 0.638 | 0.580 | 0.611 | 0.556 |
> | 10 | 0.642 | 0.624 | 0.563 | 0.595 | 0.536 |
>
>
> **ResNet18 with top-0.375 selection**
>
> | Radius | SHAP | LIME | IntGrad | MFABA | Rand |
> | ------ | ---- | ---- | ------- | ----- | ---- |
> | 0 | 1.000 | 1.000 | 1.000 | 1.000 | 1.000 |
> | 1 | 0.919 | 0.912 | 0.870 | 0.886 | 0.863 |
> | 2 | 0.882 | 0.875 | 0.811 | 0.835 | 0.802 |
> | 3 | 0.854 | 0.846 | 0.767 | 0.800 | 0.756 |
> | 4 | 0.831 | 0.824 | 0.733 | 0.771 | 0.721 |
> | 5 | 0.811 | 0.805 | 0.704 | 0.747 | 0.690 |
> | 6 | 0.794 | 0.789 | 0.678 | 0.727 | 0.661 |
> | 7 | 0.780 | 0.776 | 0.655 | 0.707 | 0.637 |
> | 8 | 0.768 | 0.763 | 0.635 | 0.690 | 0.616 |
> | 9 | 0.757 | 0.751 | 0.616 | 0.675 | 0.596 |
> | 10 | 0.747 | 0.740 | 0.598 | 0.661 | 0.579 |
>
> ## W3. Adding Suggested Citations
>
> Thank you for bringing these additional works to our attention and for providing insightful analysis. We will update our manuscript with the suggested citations and further the discussion on how they relate to our work.
>
>
> ## Addressing Questions
> **Q1.** This interpretation is correct: Please see W1 above.
>
> **Q2.** We appreciate you highlighting this work for us. While both our work and Khan et al. (2024) propose probabilistic stability metrics, they address different but complementary notions of robustness. In particular, Khan et al. measure how the explanation changes when the input is perturbed (how $\alpha, \alpha’$ vary with $x, x’$, where let $\alpha = \phi(x)$ and $\alpha’ = \phi(x’)$ for an explanation method $\phi$). In contrast, we measure how changing the explanation on some input affects the model output (i.e., $f(x\odot \alpha)$ vs. $f(x \odot \alpha’)$ for different $\alpha, \alpha’$ on the same $x$). This is a subtle but important distinction, and we will incorporate a discussion of this in our updated manuscript.
>
> **Q3.** We take $\gamma$ to be half the difference between the top-1 and top-2 class probabilities (i.e., $(p_1 - p_2)/2$, where $p_1$ and $p_2$ are the probabilities of the top-1 and top-2 classes, respectively). This ensures that even if the top-1 class probability were to drop by a value of $\gamma$, it would remain the top-1 class. Related to this, the setup in Equation 5 is intended to convey that the class probability of the perturbed prediction $f_x (\alpha’)$ should stay within a $\gamma$ radius of the original $f_x (\alpha)$. Importantly, this notion of distance to the decision boundary applies to both predictions on masked inputs and unmasked inputs. We will improve the manuscript to better describe these.

---

> > ### Comment · Reviewer_SAAu · 2025-08-01
> >
> > Thank you for your detailed response to my comments. In light of your updates to the exposition, ablation study with $k$ (which matches with intuition) and clear clarification of differences with Khan et al. (2024), I will be happy to raise my score by 1.

---

> > > ### Author Response · Authors · 2025-08-05
> > >
> > > Thank you for the encouraging assessment of our work and the score increase to a 5. We will include these ablation experiments in our manuscript revisions and expand our discussion on the suggested related work, including Khan et al.

---

### Official Review · Reviewer_LG7W · 2025-06-29

**Clarity:** 3
**Significance:** 3
**Originality:** 3
**Rating:** 5
**Confidence:** 2

**Summary:**

The paper presents a discrete probabilistic‐verification framework that offers robustness guarantees for feature attributions. By considering soft stability guarantees, the method yields tighter bounds than the hard guarantees produced by multiplicative smoothing. The results show higher stability rates across several attribution techniques.

**Questions:**

- Within the results of Figure 9, MuS with $\lambda = 0.125$ yields better results than SCA for the RoBERTa model, why is this the case? How would it perform with $\lambda = 0.1$?
- Would it be possible to expand the results to a couple of other datasets both in vision and in language?
- Would it be possible to expand the results to include an evaluation against adversarial attacks?

**Ethical Concerns:**

["NO or VERY MINOR ethics concerns only"]

**Final Justification:**

The work is technically solid and represents an improvement over prior approaches. The authors' commitment to conducting broader evaluations across additional datasets should effectively demonstrate the method's ability to generalize.

**Limitations:**

It is not clear if for lower $\lambda$ values MuS is equal to SCA.

**Paper Formatting Concerns:**

There are no concerns

**Quality:**

3

**Strengths And Weaknesses:**

### Strengths
- The paper is well written, the claims, theorems and propositions are sufficiently proved.
- The unified framework for certifying both hard and soft stability guarantees is interesting and improve on previous work.
- Enhancing probabilistic verification techniques for explainability is important to strengthen safety assurances in safety-critical systems.

### Weaknesses
- The paper offers little to no discussion on how stabilizing feature attributions might improve adversarial robustness.
- Related works in discrete randomized smoothing, e.g. [1], [2], has not been considered for comparison and/or evaluation.

[1] Hierarchical randomized smoothing
[2] Localized randomized smoothing for collective robustness certification

---

> ### Author Rebuttal · Authors · 2025-07-31
>
> We thank the reviewer for their constructive feedback and respond to their comments below.
>
> ## W1. On Stable Explanations and Adversarial Robustness
> We would like to clarify that **stability is, in fact, a form of adversarial robustness**. If one can certify that an explanation has a stability rate of 1 at some radius, then it is adversarially robust up to that perturbation radius. In this view, it is desirable for explanations to have high stability rates, preferably at 1. We will revise our exposition to make this connection more explicit.
>
> ## W2. On Discrete Randomized Smoothing
> The particular form of discrete randomness relevant to us in [1] is based on randomized maskings from [6], which is in fact the same modality that we consider. On the other hand, the discreteness in [2] is in selecting where in the input to apply continuous-valued perturbations.
>
> As such, **the listed references are either already covered or are not applicable to binary-valued feature attributions**. Moreover, random masking is the only form of discrete smoothing for which it is known how to prove guarantees on the stability rate: hard stability (adversarial robustness) from MuS in Xue et al. 2023, and the stability rate lower-bound in Theorem 4.2.
>
> Nonetheless, the suggested references do suggest interesting directions for future analysis, particularly how one might selectively apply random masking on selected subsets of the features.
>
>
> ## Addressing Questions
>
> **Q1.** The performance difference happens because we are comparing the stability of a very heavily smoothed model (MuS with $\lambda = 0.125$) with an unmodified model (SCA). In particular, MuS with $\lambda = 0.125$ means that only $1/8$th of the original features are visible to the underlying model, which heavily degrades accuracy (see Figure 8). Under such extreme trade-offs, it is not surprising that MuS’ certificates might outperform SCA, especially at small perturbation radii.
>
> On radii, we would also like to note that, in all the curves for Figure 9, MuS’ stability certificate drops off at radius $1/2\lambda$, as that is the maximum radius to which MuS can certify. In contrast, SCA can certify far beyond these radii (Figure 6).
>
> **Q2.** Yes, we can expand our results to more image and text datasets in the final manuscript. For image datasets, we are considering adding MS COCO [3] and/or Open Images V7 [4] datasets. And for text datasets, we are considering the Large Movie Review dataset [5]. If there are particular datasets the reviewer wants us to add, we are open to suggestions. Moreover, we would like to note that our dataset coverage is a superset of that used in Xue et al. (2023).
>
> **Q3.** Using adversarial attacks is an interesting suggestion. This would be useful for identifying empirical upper bounds on the hard stability radius. In fact, such experiments were conducted in Xue et al. 2023 (Figure 4), where it was observed that the MuS-certified hard stability radius is much smaller than the empirical hard stability radius.
>
> However, it is not immediately obvious how adversarial attacks on an explanation would meaningfully relate to soft stability. One preliminary idea is to see if adversarial attacks may be used to identify “sensitive” regions of an explanation, and then use this to accordingly adapt smoothing, e.g., by changing the probability that a feature is randomly masked.
>
> [3] https://cocodataset.org
>
> [4] https://storage.googleapis.com/openimages/web/index.html
>
> [5] https://huggingface.co/datasets/stanfordnlp/imdb
>
> [6] Levine et al. “Robustness Certificates for Sparse Adversarial Attacks by Randomized Ablation”. AAAI 2020.

---

> ### Comment · Reviewer_LG7W · 2025-08-03
>
> Thank you for clarifying the adversarial robustness and comparison with other discrete randomized smoothing techniques. I am increasing my score by 1 point.

---

> > ### Author Response · Authors · 2025-08-05
> >
> > Thank you for the positive evaluation and the score increase to a 5. We will improve our manuscript to better discuss the relation between soft stability and adversarial robustness, as well as future work on other discrete randomized smoothing techniques.

---

### Official Review · Reviewer_wGqH · 2025-06-30

**Clarity:** 3
**Significance:** 3
**Originality:** 3
**Rating:** 4
**Confidence:** 4

**Summary:**

Recognizing the limitations of existing stability guarantees, in this manuscript, the author(s) proposed a stability certification algorithm based on soft stability, extending prior work to the probabilistic setting. And through empirical experiments, the author(s) demonstrated the effectiveness of the proposed method. The manuscript was mainly motivated by the central idea of additive features.

---

**Questions:**

The manuscript looks interesting, theoretical results are mathematically derived and empirical experiments are performed to verify these results. After reviewing this manuscript, I have the following questions/comments, I am looking forward to the responses from the author(s). Thanks.

1. In Figure 2, the stability analysis is presented only for correctly classified instances. It would be valuable to understand how the hard and soft stability metrics behave on misclassified samples. Are there any interesting results when the algorithm is applied to incorrectly classified images? If such experiments were conducted, it would be great if the author(s) could include or discuss them, as they could provide further insight into the algorithm’s robustness and limitations.

2. In Line 76, the author(s) mentioned that two similar attributions yield the same prediction. And in the manuscript, the author(s) gave two ways including overlapping and subset/superset (i.e., additive perturbations in the manuscript) feature sets. For comparison, could you please illustrate any other ways (If there are) and cite the related work?

3. In Figure 3, the notion of similarity is central to the discussion. Could the author(s) clarify how similarity is defined? Just from the observation, or other actual metrics? Also, even if there is much overlapping, it will not be significant (maybe the cat nose patch is more important than others) if the overlapping are not key features, then it will make the stability very different also be reasonable and obvious.

4. In Line 221, about the value of $\lambda$, could we use the theoretical results obtained in the manuscript explain/guide it?

5. In Figure 5, could you please explain the cross point and the changing among yellow and green in the 3rd subfigure, yellow and red in the 4th subfigure? And why is there a sudden drop in the last subfigure? Understanding these patterns might help clarify the behavior and limitations of the proposed method under varying conditions.

6. Some other tiny issues:

(1) Please avoid repeatedly spelling out terms after their abbreviations have been introduced, such as "Stability Certification Algorithm (SCA) " is fully written multiple times in Lines 48, 108, 131, … after the abbreviation is introduced.

(2) Ensure consistency in reference formatting. There are inconsistencies such as, refs. [59] vs. [60], refs. [62] vs. [64], etc.

---

**Ethical Concerns:**

["NO or VERY MINOR ethics concerns only"]

**Final Justification:**

I will vote to accept the manuscript, i.e., maintain my original score.

**Limitations:**

Yes.

**Paper Formatting Concerns:**

For more details, please see Section "Questions".

**Quality:**

3

**Strengths And Weaknesses:**

In my opinion, the strengths and weaknesses of this manuscript are as follows:

## Strengths:

1) The proposed algorithm introduces a soft stability-based certification framework, extending prior deterministic stability guarantees to a probabilistic setting.
2) The algorithm offers stronger and more flexible stability guarantees compared to existing algorithms.
3) The author(s) implemented empirical experiments to support the effectiveness of proposed algorithm.

---

## Weaknesses:

1) While the author(s) claimed that using experiments validated the finds, the absence of accompanying code will diminish the credibility of the results and makes reproducibility more difficult.

2) Some experimental details/explanations are not clear; for example, how to determine the value of $\lambda$. For more details, please see Section ***Questions***

3) The manuscript contains some formatting inconsistencies and issues.


---

---

> ### Author Rebuttal · Authors · 2025-07-31
>
> We thank the reviewer for their thorough comments and constructive feedback. Below, we address the main points raised.
>
> ## W1. Accompanying Code
> Since we are not permitted to post any links to external pages at rebuttal and discussion time, we will release the code afterwards. We agree that reproducibility is important and will make sure that the code is easy to follow and run.
>
> ## W2. On selecting $\lambda$
> Thank you for suggesting adding a discussion on how to select $\lambda$. Since empirical results suggest that stability is mostly monotonically decreasing with $\lambda$, we may use a binary search method to quickly search over a range of candidate $\lambda$ and pick the one that achieves the best accuracy-stability trade-off.
>
> ## W3. Formatting and consistency
> Thank you for catching these inconsistencies. We will fix them in the updated manuscript (cf. Q6).
>
>
> ## Addressing Questions
>
> **Q1.** Thank you for the insightful suggestion. We have performed a preliminary ablation study using ResNet18 and 100 samples from ImageNet. For each image, we randomly sampled two masks that each selected 25% of the features: (1) a “good mask” that recovers the same prediction as the full input does and (2) a “bad mask” that gives a different prediction (i.e., misclassifies). We expect that including more features in a mask should increase the likelihood that a masked prediction recovers the same output as the full input, meaning that the “bad masks” should, on average, be less stable than the “good masks.” Indeed, we observe this in our evaluation, especially as the perturbation size (the number of added features) increases.
>
> | Radius | Good Mask | Bad Mask |
> | ------ | ------ | ------ |
> | 0 | 1.000 | 1.000|
> | 1 | 0.815 | 0.856|
> | 2 | 0.758 | 0.791|
> | 3 | 0.708 | 0.727|
> | 4 | 0.681 | 0.678|
> | 5 | 0.666 | 0.636|
> | 6 | 0.646 | 0.597|
> | 7 | 0.627 | 0.570|
> | 8 | 0.615 | 0.535|
> | 9 | 0.605 | 0.510|
> | 10 | 0.594 | 0.485|
> | 11 | 0.595 | 0.464|
> | 12 | 0.594 | 0.439|
> | 13 | 0.589 | 0.427|
> | 14 | 0.588 | 0.401|
> | 15 | 0.580 | 0.382|
> | 16 | 0.585 | 0.373|
> | 17 | 0.578 | 0.355|
> | 18 | 0.577 | 0.348|
> | 19 | 0.581 | 0.329|
> | 20 | 0.584 | 0.316|
>
>
> **Q2.** We will expand our discussion of feature attribution alignment in the updated manuscript. Similarity of attributions can also be judged by how closely their rankings agree [1], whether they preserve the sign of each contribution [1], perceptual or structural resemblance of image saliency maps [2], and higher‑level alignment in a latent or neighbourhood space [3].
>
>
> **Q3.** Our notion of similarity is based on overlapping feature sets, where 4/6 of the features are common for the two explanations. However, the reviewer brings up a valid concern about the limitation of overlap-based metrics. We will clarify and expand the discussion on this in our revised manuscript, particularly using the related works mentioned in Q3.
>
> **Q4.** Please see W2 above.
>
> **Q5.** The cross-points happen when SCA is compared to a very heavily smoothed MuS ($\lambda = 0.125, 0.250$). In such cases and at low radii ($\leq 4$), it is not unexpected that MuS might beat SCA. However, MuS' certifiable stability radius tops out at $1/2\lambda$, where note the steep drop, meaning that trend lines for MUS do not extend past radius 4 (where $\lambda = 0.125$).
>
> **Q6.** Thank you again for your attention to detail on these. We will update them accordingly in the final manuscript.
>
>
> [1] Krishna et al. “The Disagreement Problem in Explainable Machine Learning: A Practitioner’s Perspective.” 2025.
>
> [2] Szczepankiewicz et al. “Ground truth based comparison of saliency maps algorithms.” 2023.
>
> [3] Pirie et al. “AGREE: A Feature Attribution Aggregation Framework to Address Explainer Disagreements with Alignment Metrics.” 2023.

---

> ### Comment · Reviewer_wGqH · 2025-08-01
> **Thanks for the response**
>
> Thanks for the response from the author(s). I have read the replies, which clarify my confusion and concerns more or less. I will vote to accept the manuscript, i.e., maintain my original score.
>
> Regarding the statement "Release the code afterwards," I’d like to express a concern based on my past experiences — as a reviewer, I have encountered several cases where code was promised but never released or just partly, even 2–3 years after publication. I sincerely hope the author(s) of this manuscript will help restore trust in such promises by making the code available as stated — ideally with a concrete timeline and a real repository link. Thanks.
>
> I also appreciate the author(s)’ acknowledgment of the importance of reproducibility. Good luck!

---

> > ### Author Response · Authors · 2025-08-05
> >
> > Thank you for the positive feedback.
> >
> > To verify that our code and plots are present, we have sent the AC a link to an anonymized code repository. This is because, unfortunately, we are not allowed to share such links with the reviewers.
> >
> > Moreover, we will include code repository links in the public preprints of this work, which we are highly incentivized to update and polish with reviewer feedback. The expected release/update is before this coming September.

---

### Official Review · Reviewer_sbAw · 2025-07-03

**Clarity:** 4
**Significance:** 3
**Originality:** 3
**Rating:** 5
**Confidence:** 4

**Summary:**

The paper tackles certifying the robustness of feature-attribution explanations when additional features are added. It introduces soft stability, a probabilistic metric quantifying the share of perturbations that leave the prediction unchanged, and presents the Stability Certification Algorithm (SCA), a simple, model-agnostic sampling method that statistically guarantees both soft and hard stability. Leveraging new Boolean analysis, the authors prove that mild random-mask smoothing improves stability without accuracy loss, and confirm these insights on vision and language models with multiple attribution methods.

**Questions:**

- Why restrict to additive, uniformly sampled supersets $\Delta \_r$? Could SCA be extended to certify robustness to feature removals or structured perturbations (e.g., contiguous text spans or image patches)?

- Practitioners must pick $\varepsilon, \delta$ (to decide how many explanations to draw), $r$, and possibly $\lambda$. Do you have heuristics or default settings that balance certificate strength, runtime, and accuracy loss?

**Ethical Concerns:**

["NO or VERY MINOR ethics concerns only"]

**Final Justification:**

I recommend acceptance. The authors' responses adequately addressed my concerns about baseline comparisons and mask-size sensitivity analysis.

**Limitations:**

Although the limitations are not crucial, it is important that authors mention them:

- SCA needs $O\left(1 / \varepsilon^2\right)$ forward passes per input; for small $\varepsilon$ the wall-time and memory could be prohibitive

- Stability is certified only for additive feature inclusions drawn uniformly. Real edits often involve removals or structured, correlated changes, leaving those scenarios uncertified.

- Certificates are framed by ( $\varepsilon, \delta$ ) while any non-zero failure probability may be unacceptable in safety-critical settings demanding zero risk.

- The theory assumes a computable distance to the decision boundary $\gamma$ (eq. 5) while estimating even loose lower bounds is non-trivial for deep networks, so the tightness of the bounds is uncertain.

- The method certifies stability only for a single input-explanation pair. It offers no distribution-level or global guarantee akin to Lipschitz bounds.

**Paper Formatting Concerns:**

- No formatting concerns

**Quality:**

4

**Strengths And Weaknesses:**

## Strengths

- The paper is very well-written and easy to follow.
- SCA is simple and model-agnostic.
- The authors show that SCA can provably certify soft and hard stability.
- The authors show that *mild* smoothing can boost stability without the severe accuracy loss of prior certifiers, a well-known usability bottleneck.
- The authors provide a Boolean-analysis result that supplies a principled explanation for why mild smoothing helps, moving beyond purely empirical claims.
- Experiments reportedly span both vision and NLP tasks, suggesting breadth.

## Weaknesses

The main weakness is that the paper's evaluation leaves several gaps that weaken its practical impact.

### No human validation

The paper does not test whether the proposed stability score aligns with human judgements of explanation quality, leaving practical relevance unverified. I do not expect authors to address the human validation during the rebuttal, but justification for omitting it would strengthen the paper.

### Incomplete baseline coverage
The study omits empirical comparisons with other evaluations for feature attribution methods, e.g., [1 - 4]. Again, I do not expect authors to do this comparison during the rebuttal, but it would be nice to explicitly tell why these baselines are beyond the scope of the paper.

### Missing runtime evidence

SCA requires $O\left(1 / \varepsilon^2\right)$ samples (Theorem 3.1). For tight tolerances, this could be prohibitively slow, but the authors report no wall-clock time, FLOP count, or memory usage for SCA and other baselines.

### No mask-time sensitivity study

All experiments fix the explanation to the top 25 \% of features, yet attribution methods differ widely in sparsity. Without varying this threshold, we cannot know whether SCA still beats MuS at 10 \% or 40 \%, nor can practitioners learn how to pick $k$ for their own tasks.

[1] Hooker, Sara, et al. "A benchmark for interpretability methods in deep neural networks." Advances in neural information processing systems 32 (2019).

[2] Fel, Thomas, et al. "How good is your explanation? algorithmic stability measures to assess the quality of explanations for deep neural networks." Proceedings of the IEEE/CVF Winter Conference on Applications of Computer Vision. 2022.

[3] Salih, Ahmed, et al. "Investigating explainable artificial intelligence for mri-based classification of dementia: a new stability criterion for explainable methods." 2022 IEEE International Conference on Image Processing (ICIP). IEEE, 2022.

[4] Faber, Lukas, Amin K. Moghaddam, and Roger Wattenhofer. "When comparing to ground truth is wrong: On evaluating gnn explanation methods." Proceedings of the 27th ACM SIGKDD conference on knowledge discovery & data mining. 2021.

---

> ### Author Rebuttal · Authors · 2025-07-31
>
> We thank the reviewer for their thoughtful comments and positive reception. Below, we address the main points raised.
>
> ## W1. Regarding human validation
>
> Our primary focus is on certifying the stability of existing explanation methods. Exploring the alignment of stable explanations with human intuition would expand the practical relevance of this line of work and is an interesting future direction, but this is beyond the immediate, mathematically-focused scope of this paper.
>
> ## W2. On Baseline Coverage of [1-4]
> We agree that a thorough comparison to related work is crucial; however, **the suggested baselines are not directly comparable** as they measure fundamentally different properties. To clarify the difference, our work certifies the stability of a *model's prediction* when the *explanation itself* (the feature set) is perturbed. In contrast, the cited works evaluate *how the explanation changes* when the *model or its inputs* are perturbed:
> * [1] Hooker et al. use ROAR to measure feature importance by retraining models, a process that perturbs the model itself and is unrelated to prediction stability.
> * [2, 4] measure how the explanation changes when the training data or input graph is perturbed.
> * [3] measures how the feature ranking changes when the top feature is removed from consideration.
> In summary, the above methods measure the robustness of the explanation-generating process, whereas we assess the robustness (stability) of a given explanation on a particular model. The only other directly comparable prior work is the hard-stability measure from Xue et al. (2023), which we thoroughly evaluate against.
>
> ## W3. On Computational Cost and Runtime
> The reviewer raised a concern about the potential runtime of SCA, given its $O(1/\varepsilon^2)$ sample complexity. **This complexity is theoretically optimal for Monte Carlo methods without additional model assumptions.** We report sample complexity—which corresponds to the number of forward passes—as it is the **only model- and hardware-agnostic metric** for computational cost. In contrast, wall-clock time, FLOP count, or memory usage are implementation-dependent and would not provide a generalizable comparison.
>
>
> ## W4. Mask-time sensitivity study
>
> Thank you for suggesting running an ablation on the top-k features. The current choice of top-25% follows from the convention of prior work in Xue et al. 2023.
>
> As a preliminary experiment, we ran ResNet18 on 2000 samples from ImageNet with explanations taken from the top-0.125, top-0.250, and top-0.375 of the features. As the proportion of features making up the explanation mask increases, we see that the stability increases, which is expected.
>
> Importantly, MuS can still only certify up to a radius of at most $1/2\lambda$ and even this requires a heavily smoothed model (e.g., $\lambda = 0.125$ means that on average, only $1/8$th of the original features are visible to the underlying model). Thus, even if MuS “beats” SCA (certain plots in Figure 9), this can only happen under very heavily smoothed conditions, and even then, only at low radii.
>
>
> **ResNet18 with top-0.125 selection**
>
> | Radius | SHAP | LIME | IntGrad | MFABA | Rand |
> | ------ | ---- | ---- | ------- | ----- | ---- |
> | 0 | 1.000 | 1.000 | 1.000 | 1.000 | 1.000 |
> | 1 | 0.817 | 0.796 | 0.819 | 0.807 | 0.834 |
> | 2 | 0.734 | 0.705 | 0.742 | 0.726 | 0.756 |
> | 3 | 0.674 | 0.642 | 0.685 | 0.668 | 0.702 |
> | 4 | 0.625 | 0.595 | 0.640 | 0.623 | 0.656 |
> | 5 | 0.584 | 0.554 | 0.601 | 0.587 | 0.617 |
> | 6 | 0.552 | 0.521 | 0.567 | 0.554 | 0.583 |
> | 7 | 0.524 | 0.494 | 0.536 | 0.526 | 0.550 |
> | 8 | 0.501 | 0.469 | 0.507 | 0.503 | 0.523 |
> | 9 | 0.480 | 0.450 | 0.483 | 0.480 | 0.495 |
> | 10 | 0.461 | 0.433 | 0.458 | 0.458 | 0.473 |
>
>
> **Resnet18 with top-0.250 selection**
>
> | Radius | SHAP | LIME | IntGrad | MFABA | Rand |
> | ------ | ---- | ---- | ------- | ----- | ---- |
> | 0 | 1.000 | 1.000 | 1.000 | 1.000 | 1.000 |
> | 1 | 0.885 | 0.871 | 0.848 | 0.866 | 0.845 |
> | 2 | 0.830 | 0.813 | 0.783 | 0.806 | 0.778 |
> | 3 | 0.790 | 0.774 | 0.740 | 0.761 | 0.728 |
> | 4 | 0.757 | 0.740 | 0.701 | 0.727 | 0.691 |
> | 5 | 0.731 | 0.713 | 0.673 | 0.697 | 0.655 |
> | 6 | 0.707 | 0.691 | 0.647 | 0.672 | 0.627 |
> | 7 | 0.687 | 0.670 | 0.623 | 0.649 | 0.600 |
> | 8 | 0.670 | 0.653 | 0.601 | 0.629 | 0.578 |
> | 9 | 0.655 | 0.638 | 0.580 | 0.611 | 0.556 |
> | 10 | 0.642 | 0.624 | 0.563 | 0.595 | 0.536 |
>
>
> **ResNet18 with top-0.375 selection**
>
> | Radius | SHAP | LIME | IntGrad | MFABA | Rand |
> | ------ | ---- | ---- | ------- | ----- | ---- |
> | 0 | 1.000 | 1.000 | 1.000 | 1.000 | 1.000 |
> | 1 | 0.919 | 0.912 | 0.870 | 0.886 | 0.863 |
> | 2 | 0.882 | 0.875 | 0.811 | 0.835 | 0.802 |
> | 3 | 0.854 | 0.846 | 0.767 | 0.800 | 0.756 |
> | 4 | 0.831 | 0.824 | 0.733 | 0.771 | 0.721 |
> | 5 | 0.811 | 0.805 | 0.704 | 0.747 | 0.690 |
> | 6 | 0.794 | 0.789 | 0.678 | 0.727 | 0.661 |
> | 7 | 0.780 | 0.776 | 0.655 | 0.707 | 0.637 |
> | 8 | 0.768 | 0.763 | 0.635 | 0.690 | 0.616 |
> | 9 | 0.757 | 0.751 | 0.616 | 0.675 | 0.596 |
> | 10 | 0.747 | 0.740 | 0.598 | 0.661 | 0.579 |
>
>
> ## Addressing Questions
> Moreover, we address the specific questions raised below:
>
> **Q1.** We focused on additive subsets to provide a direct comparison with the perturbation style described in Xue et al. (2023). However, our framework can easily be extended. Certifying robustness to feature removals is achievable with a simple modification to our sampling algorithm in Section 3.2 (i.e., sampling from the '1's in the feature mask instead of the '0's). Structured perturbations would require a more complex sampling strategy that respects the desired structure (e.g., sampling contiguous blocks), which we believe is a promising direction for future work. We will clarify this in the paper.
>
> **Q2.** Yes, the reviewer is correct that one must select parameters. While the optimal choice of parameters **depends on the application**, we offer the following guidance:
>  * Practitioner-defined Parameters ($\varepsilon, \delta, r$): These parameters are standard sample complexity notions that are typically set by the user based on what is meaningful for their specific use case and/or are governed by the runtime the user is willing to pay for. Specifically, $\varepsilon$ (tolerance) and $\delta$ (confidence) are to be set at the desired statistical guarantees, while $r$ (perturbation size) is a user and task-specific criterion.
> * Tunable Parameter ($\lambda$): This can be tuned like a standard hyperparameter. We recommend performing a simple sweep on a validation set to find a value that maximizes stability without a significant drop in accuracy for the whole dataset.
>
> We also appreciate the reviewer for taking the time to enumerate limitations, and we will update our manuscript to reflect them.

---

> > ### Comment · Reviewer_sbAw · 2025-08-02
> >
> > Thank you for your clarifications. I recommend paper acceptance.
> >
> > For the camera-ready version, please explicitly encourage the future work dedicated to evaluating whether the proposed stability measure aligns with human judgements of explanation quality. Without such validation, it remains unclear whether the stability measure is meaningful in practice. This raises the risk that the current notion of stability may not correspond to human-interpretable or useful explanations.
> >
> > Furthermore, please include empirically measured runtime costs (e.g., FLOPs, latency) for both SCA and the baseline from Xue et al., assuming a fixed architecture and compute environment. I read your answer to that request, but I still believe this would help clarify the practical trade-offs of your approach.

---

> > > ### Author Response · Authors · 2025-08-05
> > >
> > > Thank you for your positive reception. We will accordingly update our paper to encourage human evaluations in future work, as well as include wall-clock times in our updated experiments.

---

### Decision · Program_Chairs · 2025-09-17

**Decision:**

Accept (poster)

**Comment:**

This paper introduces the Stability Certification Algorithm (SCA), which provides probabilistic guarantees for explanation robustness through the notion of soft stability.

Reviewers agreed that the work is technically solid and makes a meaningful contribution to interpretability by offering a more flexible and practical certification framework than prior methods. Theoretical results are clearly presented and supported with Boolean function analysis, and experiments on both vision and language tasks demonstrate the algorithm’s versatility.

Limitations include the lack of human studies to test whether stability aligns with human judgments, limited runtime and efficiency analysis, and incomplete baseline comparisons. The authors justified these choices and added clarifications and ablations during the rebuttal.

Overall, reviewers agreed that the paper is worthy of acceptance. The framework is simple, well-grounded, and improves upon existing certification techniques. Additional experiments and clarifications during the rebuttal also addressed some key concerns.

The authors should make sure to update their paper based on the discussions including releasing code, encouraging future work for evaluating stability compared to human judgments, empirical runtime costs, and discussion w.r.t. adversarial robustness.